# Axial de-scanning using remote focusing in the detection arm of light-sheet microscopy

Hassan Dibaji[1], Ali Kazemi Nasaban Shotorban [1], Rachel M. Grattan[2,3], Shayna Lucero[2,3], David J. Schodt [1], Keith A. Lidke [1,2], Jonathan Petruccelli[4], Diane S. Lidke[2,3], Sheng Liu [1] & Tonmoy Chakraborty [1,2] ✉

Rapid, high-resolution volumetric imaging without moving heavy objectives or disturbing delicate samples remains challenging. Pupil-matched remote focusing offers a promising solution for high NA systems, but the fluorescence signal's incoherent and unpolarized nature complicates its application. Thus, remote focusing is mainly used in the illumination arm with polarized laser light to improve optical coupling. Here, we introduce a novel optical design that can de-scan the axial focus movement in the detection arm of a microscope. Our method splits the fluorescence signal into S and P-polarized light, lets them pass through the remote focusing module separately, and combines them with the camera. This allows us to use only one focusing element to perform aberration-free, multi-color, volumetric imaging without (a) compromising the fluorescent signal and (b) needing to perform sample/detection-objective translation. We demonstrate the capabilities of this scheme by acquiring fast dual-color 4D (3D space + time) image stacks with an axial range of 70 μm and camera-limited acquisition speed. Owing to its general nature, we believe this technique will find its application in many other microscopy techniques that currently use an adjustable Z-stage to carry out volumetric imaging, such as confocal, 2-photon, and light sheet variants.

Fast 3D positioning or scanning of an optical system's focal point or focal plane has the potential to transform many areas of biophotonics, especially those that require studying the complex dynamics of living organisms. Processes like investigation of neuronal activities of the brain, blood flow in the heart, and cell signaling require high-speed volumetric imaging[1–3]. However, volumetric imaging requires an axial scan either through the translation of the sample or the detection objective lens (Fig. 1a). Such axial translations result in imaging modalities that are often slow, with speeds limited to a few hundred Hz[4–6]. Additionally, with fragile samples, such as an expanded sample in hydrogel[7], fast movements of the sample stage may agitate the sample and induce distortions when collecting volumetric images. To avoid the slow translation of bulky objectives or the sample stages, several

attempts, employing variable-focus (vari-focus) lenses, mechanical mirrors, and acousto-optic modulators, have been proposed to refocus the light for 3D imaging. However, they all suffer from unacceptable aberrations introduced by the focusing elements. A large category of those techniques utilizes different types of tunable lenses such as ferroelectric liquid crystal (LC), acoustic waves (TAG lens), and acoustic optics modulators (AOM)[8] to achieve fast focal shifts (~ 1 kHz). Ferroelectric LC and TAG lenses introduce a focal shift by varying the gradient of the refractive index of the liquid medium, however, the generated phase variation only approximates the defocus phase, leading to increased spherical aberration at large focal shifts[9–11]. AOM-based vari-focus techniques on the other hand use two AOMs with counterpropagating acoustic waves to cancel out the transverse scan

[1]Department of Physics and Astronomy, University of New Mexico, Albuquerque, NM, USA. [2]Comprehensive Cancer Center, University of New Mexico Health Sciences Center, Albuquerque, NM, USA. [3]Department of Pathology, University of New Mexico Health Science Center, Albuquerque, NM, USA. [4]Department of Physics, University at Albany–State University of NewYork, Albany, NY, USA. ✉e-mail: tchakraborty@unm.edu

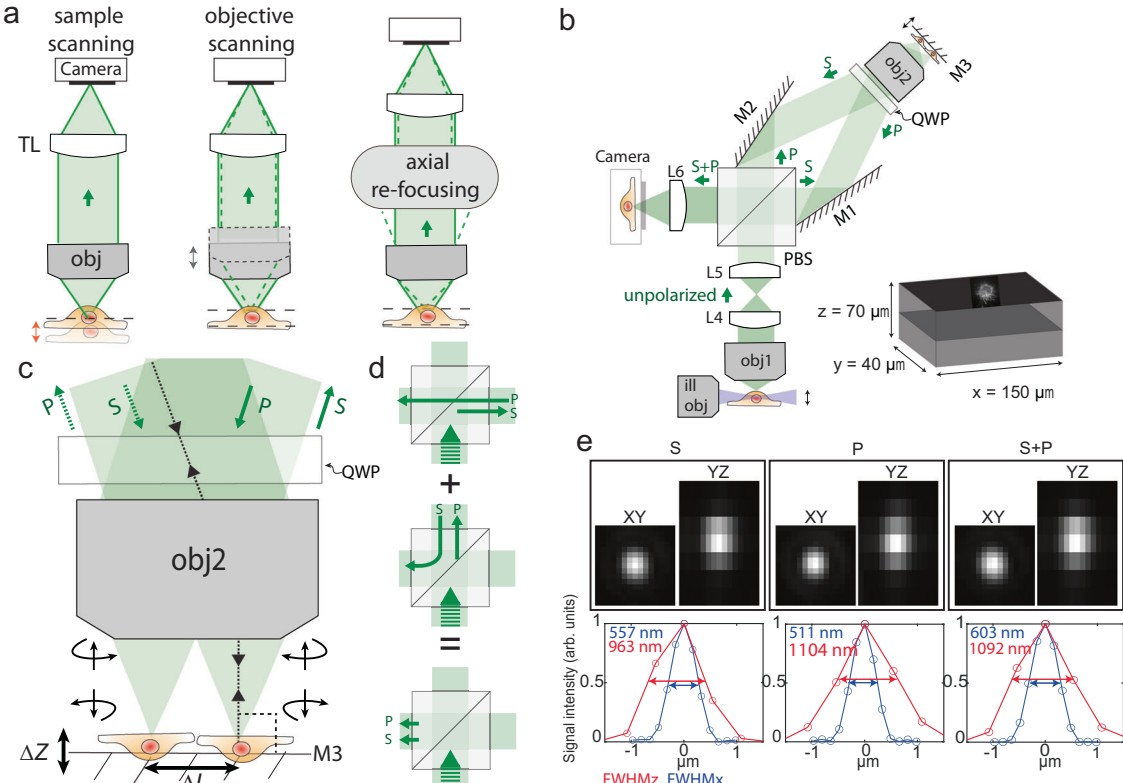

**Fig. 1 | Schematic diagram of a remote focusing system implemented in light-sheet microscopy and its performance. a** Three different modalities to acquire volumetric imaging of the sample along the focus direction. Either the sample or objective lens (obj) can be moved for axial re-focusing. Alternatively, both the sample and objective lens can remain stationary by using a remote focusing system. **b** Implementation of the remote-focusing system on the detection arm of the light sheet microscope. Here, objective lenses 1 and 2 are pupil-matched through lenses L4 and L5. Two tilted mirrors M1 and M2 are utilized to direct both S and P-polarized beams toward Objective lens 2 (obj2) and then combine the reflected beams from mirror M3 to create an image by S and P-polarized beams onto the camera by focusing through the lens (L6). The mirror M3 is attached to the linear focus actuator (LFA), moving back and forth to scan the sample in the *Z*-direction to acquire a 3D image. In the illumination arm, the generated light sheet by a cylindrical lens is translated by a galvanometric scan mirror (GSM) along the detection arm. To focus the detection path on the plane of the light sheet, the synchronization of GSM and LFA is carried out by sawtooth signals. Simultaneous dual-channel imaging of the cell is achieved in 40 μm × 150 μm FOV over 70 μm in the *Z*-direction. **c** The polarization state of the incoming beams changes after reflection from mirror M3 (S to P, and P to S). **d** The reflected beams from mirror M3 have a different polarization state compared to the incoming beams; therefore, they exit from a different side of the PBS than the incoming beams. **e** Example bead measurements of 200 nm beads formed by S, P, and S + P polarized beams. The microscope performs at the diffraction limit resolution of 570 ± 26 nm and 666 ± 44 nm for S + P in the *X* and *Y* directions respectively and achieves an axial resolution of 924 ± 14 nm. The reported number indicates mean±s.t.d (standard deviation) over measurements from 20 beads.

but can only achieve focus shift in one dimension (acting as a cylindrical lens)[12,13].

Adaptive optics-based vari-focus techniques overcome these limitations through accurate wavefront control using either a spatial light modulator (SLM) or a deformable mirror (DM), which can achieve a response rate of ~1 kHz and 20 kHz respectively. However, SLMs are polarization and wavelength-dependent and cannot model a continuous wavefront of the defocus phase due to its limited phase modulation depth. Large phase shifts are generated through multiple phase-wrapping of $2\pi$. With finite fly-back at the phase-wrapping borders, part of the incident light is not correctly modulated and results in decreased intensity at the focus[14]. DMs are not polarization and wavelength-dependent and can model a continuous defocus wavefront. However, the axial scan range of a DM is limited by the stroke length of the DM actuators. For example, for an objective with a numerical aperture (NA) of 0.8, the maximum axial scan range that DM-based techniques can generate is ~100 μm[15,16]. Furthermore, using DM for focus control requires accurate alignment and complicated calibration of the DM to reduce the aberrations caused by imaging samples out of the nominal focal plane of the objective[9].

Unlike the adaptive optics or DM-based approaches that require correcting the defocus plane-by-plane, pupil-matched remote focusing (pmRF), pioneered by Botcherby et al.[17,18], instantaneously corrects defocus across 3D volumes for high-NA optics thereby conserving the microscope's temporal bandwidth[17–27]. In addition, because pmRF allows precise mapping of the wavefront coupled into the back-pupil of the objective, where the angular magnification is unity, such techniques have been routinely used to carry out aberration-free high-quality axial focus control[17–27]. In pmRF techniques, a fast axial scan is achieved by the translation of a small mirror in front of the remote objective using a focus actuator[19,20,24] or by a lateral scan of a galvo mirror in combination with a step or tilted mirror at the remote objective[28]. Because of the fast response time of the focus actuator or the galvo mirror, an axial scan rate of 1–5 kHz or 12 kHz can be achieved respectively. However, current pmRF techniques for focus control are primarily limited to the illumination path. This is because pmRF uses the concept of optical isolators[29], where the polarization of the returning beam is rotated orthogonally to the incoming beam so that it can be separated from the incoming beam at the polarized beam splitter (PBS) (Supplementary Fig. 1a). This configuration ensures minimum light loss through the pmRF module but requires the incoming beam to be polarized, which is why this method is primarily used in the illumination arm where illumination laser light is usually polarized in nature and its manipulation through the optical isolator

can be easily done. In the detection arm, however, due to the fluorescence anisotropy[25], the emitted fluorescence may be partially polarized in nature. To the best of our knowledge, using purely linear optical elements (like lenses, PBSs, mirrors, waveplates, etc.), lossless conversion of unpolarized light into either S or P polarization state is not yet possible[30,31] (Supplementary Note 1). Therefore, achieving 100% efficiency in transmitting fluorescent light in and out through an optical isolator, which is used in folded pmRF geometry, is not feasible. As a result, microscopes that use optical isolator based pmRF to carry out axial scanning may incur up to 50% light loss due to one state of the polarized light being discarded after the PBS[17,22,25] (Supplementary Fig. 1a). It should be noted that Botcherby's original refocusing design involving a 3rd objective could collect the entire fluorescence signal, however this design warrants translation of bulky objectives which would slow down the axial scan speed[17].

A straightforward method to mitigate this problem is to have another copy of the pmRF module at the unused port of the PBS (Supplementary Fig. 1b) to collect the other half of the fluorescent light. However, this would require precise synchronization of two linear focus actuators (LFA), which is not only a difficult task at high speeds but also will be expensive since this method warrants two such LFAs. In this article, we present an optical design that overcomes these problems and presents a modular setup that can perform remote focusing on the detection arm of a fluorescent microscope without incurring polarization-induced losses. When attached to a light-sheet microscope, this technique allows optical refocusing without requiring the movement of the sample, or the detection objective (Fig. 1b and Supplementary Fig. 1c). As a result, the microscope can acquire 3D volumetric data limited by camera speed. This technique is applicable to many other microscopy techniques that currently use an adjustable *Z*-stage to carry out volumetric imaging such as confocal, 2-photon, and light sheet variants.

## Results

### Concept and microscope layout

Optical axial-refocusing: Our refocusing unit is shown in Fig. 1b. Here, the water immersion detection objective (Obj1) is pupil matched to a second air objective (Obj2) through two intermediate lenses following the original design by Botcherby et al.[17,18]. However, unlike traditional refocusing geometry, we split the collected unpolarized fluorescence into S and P-polarized light using a polarizing beam splitter cube (PBS) in the infinity space of Obj2. The generated orthogonal paths are then projected onto Obj2 using two angled mirrors M1 and M2. Because of this angular launch in infinity space, Obj2 forms two distinct laterally shifted images at its nominal focal plane. A small mirror placed on an LFA reflects the light back through the path it came from where a quarter wave plate (QWP) converts the S-polarized light to P on its way back (and P-polarized light to S) after being reflected from the mirror (Fig. 1c). When the returning light (in each arm) reaches the PBS, it now acts as an optical valve where the S path (which was initially P) gets reflected while the P-polarized light (which was initially S) gets transmitted by the PBS. As a result, both S and P polarized light exits the PBS through the fourth and unused face of the PBS cube (Fig. 1d). This light after passing through a lens (L6) forms identical images, one with S and another with P, at the sCMOS camera. A precise alignment using mirrors M1 and M2 overlays the two images, thereby resulting in a combined image by simply an incoherent addition without any interference artifacts.

There are a few important design considerations that need to be considered for our de-scanning setup. Firstly, it is essential that mirror M3 consistently translates along the optical axis of Obj2 without any angular deviation during the LFA's oscillatory motion. This prevents any unwanted focal shifts between the S and P paths, ensuring that the resulting image from both S and P polarizations remains focused on the camera at the same time. This arrangement ensures that both

beams return through their incoming paths, resulting in easier alignment for overlaying the final images formed by the S and P-polarized beams.

Secondly, it is advantageous that θ (angle between S and P polarized beam hitting the Obj2) (Supplementary Fig. 2) be as small as possible because this directly controls the distance between the two focal points at M3 (depicted by ΔL in Fig. 1c). A smaller ΔL ensures: (1) a smaller mirror could be utilized to carry out the remote-focusing, reducing the inertial load on the LFA, and enhancing its efficiency; (2) the alignment becomes less sensitive to tip-tilt misalignment of M3; and (3) for a particular objective the two images are more towards the center of the field of view (FOV), which may help reduce field-dependent aberrations and improve the collection efficiency.

Thirdly, there exists an inverse relationship between the angle θ and the distance between Obj2 and the PBS (inset of Supplementary Fig. 2). Therefore, this gives us an option: either adhere to the 4 f system or minimize θ. We found that for our matching objectives, Obj1 and Obj2 the 4f system (with matching lenses L4 and L5) resulted in a θ of 20° (inset of Supplementary Fig. 2). However, operating in this range poses a risk as it is challenging to ensure that both reflected beams are entirely captured by Obj2. Hence, there is a balance between adhering to the 4 f system and minimizing the angle θ. We found that our current design still allows us to achieve resolution comparable to that of diffraction-limited systems (Fig. 1e), by compromising the 4f arrangement to minimize θ.

Finally, because we generated two identical images on the camera using S and P-polarized light, it was crucial to overlay these images with precision higher than the diffraction-limited resolution to produce the final image. To do this, we developed a cross-correlation-based algorithm that quantifies the shift between overlayed S and P images in real-time with sub-pixel accuracy, allowing interactive adjustment of the mirrors M1 and M2 during system alignment.

Implementation in a light-sheet system: In order to test the performance of our design, we implemented this setup into the detection arm of a light-sheet microscope with orthogonal illumination and detection objectives. The system layout is shown in (Fig. 1b and Supplementary Fig. 2). The sample is illuminated by a sheet of light generated with a cylindrical lens in the illumination arm, and the emitted fluorescence from the sample is collected by the detection objective lens, which is set orthogonal to the illumination objective lens to capture 2D information from the sample. A galvanometric scan mirror (GSM) in the illumination arm translates the light sheet in the *Z*-direction. Because the position of the LFA in the detection arm determines the focal plane of the detection objective lens, we synchronized the GSM and LFA with the sawtooth signal to ensure that the detection path is always focused on the plane of the light-sheet (Supplementary Fig. 3). This allowed us to carry out volumetric imaging by acquiring a sequence of images from different focal planes. The LFA moves back and forth rapidly, synchronized with the movement of the GSM enabling us to quickly collect 3D image stacks.

The optical correction of defocus in our high-NA microscope allowed fast de-scanning of a 3D volume over an axial range of ~70 μm at speeds limited primarily by the camera framerate ( ~ in our case 799 camera frames/s at 2304 × 256 pixels using Hamamatsu Orca-fusion BT). We employed a dual-color imaging strategy by partitioning the FOV, enabling the simultaneous capture of two distinct fluorescent labels within each slice without sacrificing imaging speed. To do this, we used a pair of dichroic mirrors to separate the emitted wavelengths from the two labels into side-by-side dual-color images (Supplementary Fig. 2). Once acquired, these separate image sets are then precisely registered and merged to generate 4D (*X, Y, Z,* and *λ*) stacks. By sequentially capturing 4D stacks, we generated 5D (*X, Y, Z, λ,* and time) datasets that allowed us to track the dynamic behavior of biological processes. It is important to note that our setup is wavelength-

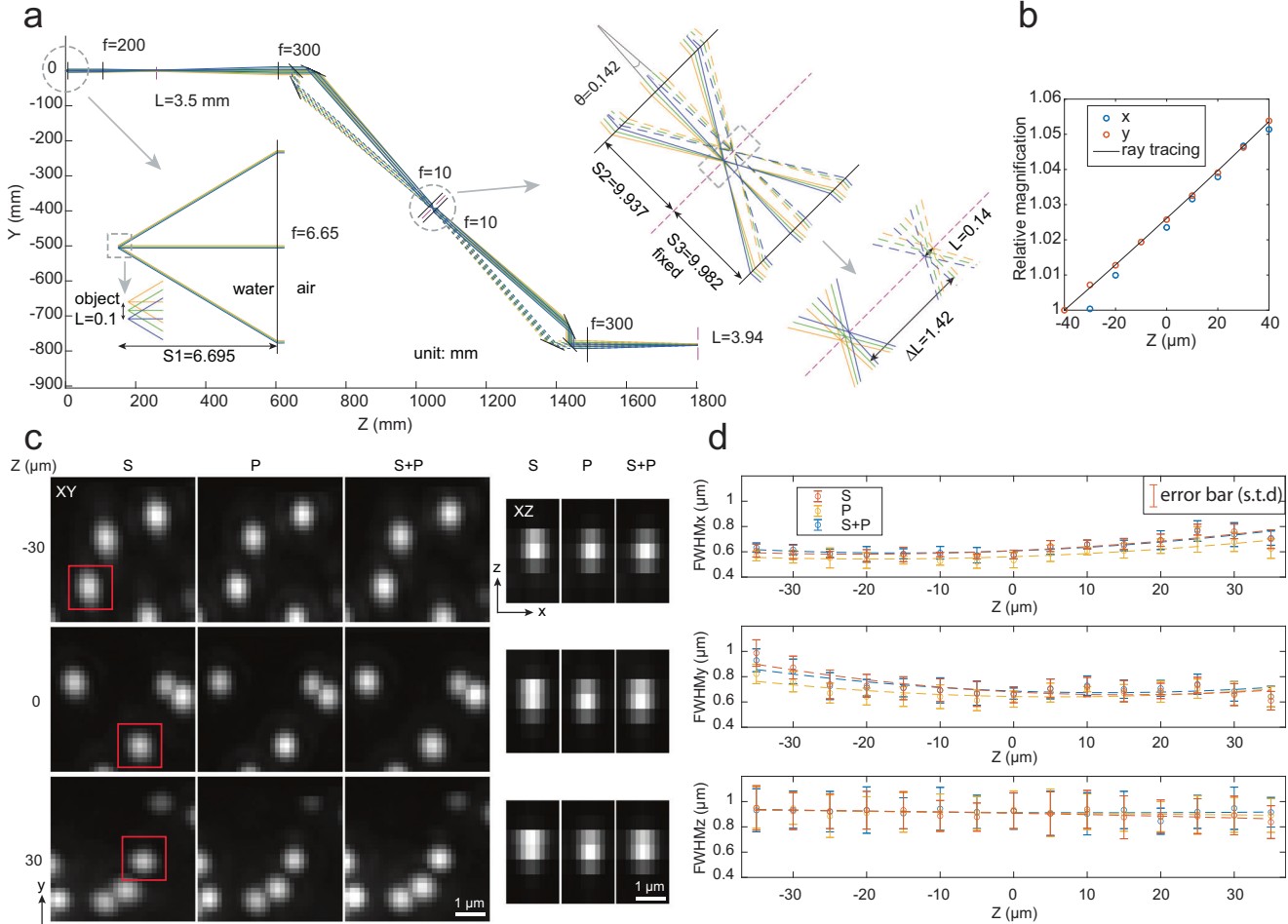

**Fig. 2 | Ray tracing of the setup and resolution assessment. a** Ray tracing of the detection path. L: image size, f: effective focal length, S: image or object position relative to the lens, unit: mm. **b** Calibration of lateral magnification at various object positions, a target illuminated by a white light LED is imaged for magnification measurement. **c** Maximum intensity projections of data acquired on 200 nm beads from 10 slices spaced 500 nm in the Z-direction. The images show orthogonal views of the MIPs across scan range for S, P, and S + P. The elongated PSF in the Z direction exhibits less resolution in the axial direction controlled by the light sheet waist. **d** The FWHM of the 200 nm beads ($n = 20$) in the lateral and axial directions over the scan range. The minimum lateral resolution, 570 nm, occurs at the center of the scan range and increases by moving away from the center. These plots show nearly a constant axial resolution of ~920 nm over the axial scan range. The microscope functions in the scan range of 70 μm. The reported number indicates mean±s.t.d (standard deviation) over measurements from 20 beads.

independent, an attribute not feasible with technologies like diffractive tunable lenses or spatial light modulators.

## Characterization of the optical system

To understand the image formation of the proposed setup, we simulated the ray tracing of the detection path (Fig. 2a). The ray tracing assumes all rays satisfy paraxial approximation and all lenses are thin lenses. The detection objective is a water immersion objective, we calculated its effective focal length as $f_{obj} = f_{Tube}n/M_{obj}$, where $f_{Tube}$ is the focal length of the designed tube lens, $M_{obj}$ is the magnification of the objective, and $n$ is the refractive index of water. Here we have $f_{obj}$ equal to 6.65 mm. The pmRF module (from the beam splitter to LFA) is modeled two times to simulate the forward and backward transmission through the module. The LFA is omitted from the simulation, instead, we change the distance between the two copies of obj2 so that the distance ($S_3$) of the image plane (magenta dash line) to the second obj2 remains constant. We simulated with an object of 100 μm, the image size after obj2 is ~140 μm, indicating a lateral magnification of 1.4, which is close to the requirement of perfect imaging with $M_{lateral} = n_{water}/n_{air} = 1.33$. This 5% magnification mismatch is limited by the geometry of the pmRF module: the separation ($\Delta L$) of the S and P-images formed by obj2 is approximate to $\Delta L = f_{obj2}\theta$, where $f_{obj2} = 10$

mm is the effective focal length of obj2 and $\theta$ is the angle between the S and P-polarized rays meeting at the obj2. The larger the $\Delta L$, the larger the aberration introduced by obj2. To reduce $\Delta L$, obj2 is located ~500 mm from the PBS, therefore, the pmRF module is no longer an exact 4 f system, and the magnification, $M_{lateral}$, varies with the axial position of the object. Furthermore, the beam path from the detection objective (Obj1) to the tube lens (L4) is also not a 4 f system, where the tube lens (L4) is ~100 mm away from the detection objective (Obj1). The combination of the two non-4f systems can partially reduce the axial dependence of the magnification. Figure 2b shows the change of the lateral magnification with respect to the galvo position (the axial position of the light sheet) from both ray tracing and the experimental data. There is a ~5% magnification change over an axial range of 80 μm.

To quantify the performance of the proposed scheme, we used full width at half maximum (FWHM) measurements of the 3D point spread function (PSF) to validate that the incoherent addition of S and P images was not compromising the resolution. To do this, we measured the PSF of each polarization component individually and compared it with the PSF of the unified S + P image. As illustrated in Fig. 1e, both the S and P-polarized images rendered onto the camera exhibit comparable FHWM, resulting in an equivalent resolution for the combined S + P image. Further quantification involving 20 randomly

chosen beads reveals that the microscope achieved resolutions of 570 ± 26 nm and 666 ± 44 nm laterally in $X$ and $Y$ directions respectively and 924 ± 14 nm axially in $Z$ direction. These measurements were performed by calculating the maximum intensity projection (MIP) of 10 slices around the nominal focal plane, with a step size of 500 nm.

To evaluate the performance of the de-scanning system, we imaged 3D volumes of 200 nm beads embedded in a 2 % agarose cube across the scan range and accessed the quality of the generated PSFs. Figure 2c displays the beads' MIPs across the scan range. For comparison we show S, P, and combined S + P images, at the beginning (-30 μm), center (0 μm) and end ( + 30 μm) of the scan range. These MIPs are generated from ten consecutive axial slices with a step size of 500 nm, using raw, unprocessed images. We found that our remote focusing setup demonstrated close to diffraction-limited performance over a scan range of ~70 μm. As evident from the 'S' and 'P' images, the quality of the beads visually appears similar across the entire scan range, thereby resulting in a similar 'S + P' image. In the axial direction (the $YZ$ view), the PSF widths are limited by the waist of the Gaussian light sheet[20,32–36] (beads from red boxes in the $XY$ view), which was determined by the tradeoff that exists between the FOV and axial resolution. We found that to image an entire cell, we needed a light sheet that would generate an FOV of ~8 μm (Supplementary Fig. 4). As a result, we reduced the NA of the illumination objective and chose a light sheet that gives a minimum axial resolution of ~ 850 nm (quantified by FWHMz of bead images).

Figure 2d displays the measured FWHMs from 200 nm beads for S, P, and S + P polarized images in the lateral ($XY$) and axial ($Z$) directions across the entire scan range. 20 randomly chosen beads were selected at each Z position. The selection criteria for the beads at each $Z$ position include ensuring that they are well-separated to prevent overlap in measurements and sufficiently bright for accurate resolution assessment. It is performed by MATLAB code, where beads were chosen by cross-correlation between the maximum intensity projection of the PSF model along $Z$ and beads in each of 10 slices. The circles indicate the mean resolution measurements obtained from the beads at each $Z$ position, providing a point of reference for the average system performance at different depths. The error bars show the standard deviation from the mean resolution at each $Z$ position, quantifying the spread of measurements and thus the consistency of the system's performance across the field of view. The dotted line represents the polynomial curve fitting over average data (circles). It serves as a trend to show resolution changes over the scanning range. The figure shows a lateral FWHM of 570 ± 26 nm and 666 ± 44 nm in $X$ and $Y$ directions at the center of the scan range respectively, which slowly increases as the beads move away from the nominal focal plane. This can be attributed to residue index mismatch aberrations that were not corrected by the remote focusing system[22]. Additionally, we found that the S polarization path suffered more in lateral resolution compared to the P polarization path, and the trend is different along the $X$ and $Y$ directions. This asymmetric FWHMs ($X$-$Y$) across the scan range ($Z$) and the discrepancy between S and P paths are likely due to field-dependent aberrations from Obj2, where the S and P images were formed at different field points of Obj2 (Fig. 1b). Furthermore, our microscope shows a nearly constant axial FWHM of ~ 920 nm over the entire scan range as the axial resolution is mainly determined by the light-sheet waist.

### Fast 3D live cell imaging

As a first demonstration of the 3D cellular imaging capabilities, we monitored the 3D motion of secretory granules in living mast cells. Mast cells possess distinct secretory granules that contain the mediators of the allergic response and are released upon mast cell activation by allergen[37]. These granules are distributed across the cytosol and have been shown to undergo both Brownian diffusion and directed motion[37]. Upon activation of the membrane receptor, FcεRI, via

crosslinking by multivalent antigen[38,39], the granules undergo increased directed motion that moves them to the plasma membrane where they will fuse and release mediators that regulate allergic responses[37,40].

We applied the developed system for dual-color, volumetric imaging of live cells and tracked the 3D motion of green fluorescent protein-labeled Fas ligand (GFP-FasL) loaded secretory granules in the cytosol of RBL-2H3 mast cells[37]. IgE-bound FcεRI was simultaneously imaged by the addition of anti-DNP IgE-CF640R. With the addition of the antigen-mimic, DNP-conjugated to BSA (DNP-BSA), FcεRI aggregates and undergoes endocytosis as seen in Fig. 3a. During data acquisition, the light sheet is parallel to the $XY$ plane and scans along the $Z$ direction. Within the light-sheet region, the $XY$ and $XZ$ maximum intensity projections (Fig. 3a) of the cell image show GFP-FasL granules in three dimensions. The cells were imaged at ~ 0.6 volumes (80 × 15 × 40 μm³ in $XYZ$) per second for 80 volumes, for a total imaging time of ~2 min (Fig. 3a–d). To quantify the granule dynamics, isolated granules were identified and tracked in 3D using the U-track3D software[41]. We calculated the mean square displacement (MSD) of each trajectory over time and extracted the diffusion coefficient, $D$, and velocity, $v$, by fitting the MSD curve with $MSD(t) = 6Dt^2 + v^2t^2 + o$, where $o$ is an offset related to localization and tracking uncertainties[42,43] (Fig. 3c, d). We found that most / granules undergo Brownian Diffusion and a few exhibited directed motion, consistent with granules being transported along the microtubules (Fig. 3a, b)[37]. The measured transport velocities of the two trajectories indicated in Fig. 3a, b are ~0.1 μm/s, consistent with previous work that performed tracking in 2D[37].

To test the limits of the system in terms of speed, we set out to image Brownian motion on the microscopic level. For this, we stressed the cells by incubating them in Hank's balanced salt solution (HBSS) (Method) at room temperature for over 1 h, which induced cell blebbing. This also caused more rapid diffusion of the granules that we were able to capture using an imaging speed of ~8.3 volumes/s for 80 volumes for a total time of 10 s. With this imaging speed, we retained good signal-to-noise and the ability to track the 3D motion of individual granules (Fig. 3e–g). Under these non-physiological conditions, average granule diffusion was increased by ~41 times (Fig. 3h). Two tracks shown in Fig. 3e have diffusion coefficients of 0.41 μm²/s and 0.64 μm²/s.

## Discussion

In this work, we developed an axial scanning module in the detection path of a light-sheet microscope utilizing the pmRF technique proposed by Botcherby et al.[17,18]. While inheriting all the benefits from the pmRF technique, such as fast scanning and all-optical aberration compensation (no wavefront control element), our design overcomes a critical limitation of the original pmRF technique, as in the loss of 50% of the emitted fluorescence in the detection path[22,25,44]. Here, we engineered an optical design, where we split the emitted fluorescence into S and P-polarized light to carry out remote focusing and then seamlessly combine them to achieve minimum light loss. We demonstrated our implementation of the developed scanning module through a light-sheet microscope with two orthogonally arranged objectives. We can perform simultaneous two-color imaging at 8.3 volumes (80 × 15 × 40 μm³ in $XYZ$) per second with a lateral resolution of 394 nm and an axial resolution of 650 nm (after deconvolution). As our method is fully optical, the imaging speed scales with advancements in LFA technology and camera acquisition speed.

The S and P polarized beams are directed at an oblique angle into the remote objective (Fig. 1b). This angled approach creates two separate images at the mirror attached to the LFA (M3). However, there are limitations to this angular arrangement. The two images formed away from the optical axis are prone to aberrations. To reduce the image separation, the remote objective must be positioned further from the PBS to reduce the incident angles of S and P-polarized lights.

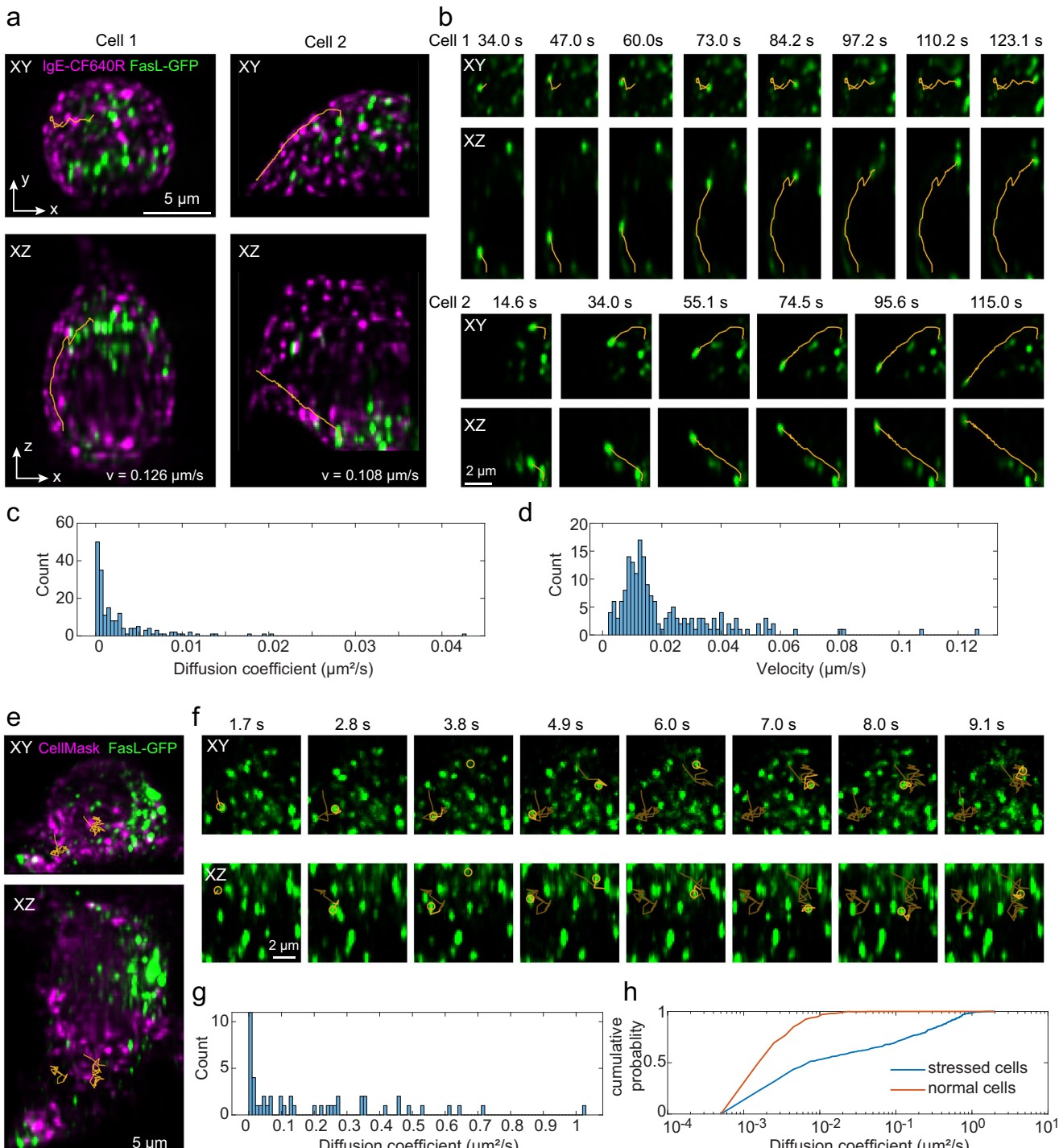

**Fig. 3 | Dual-color volumetric imaging of live RBL cells. a–d** Dual-color volumetric imaging of granule motions in a live RBL-2H3 GFP-FasL cell, where the cell membrane is labeled with IgE-CF640R and granules contain GFP-FasL, at an imaging speed of ~ 0.6 volumes ($80 \times 15 \times 40\,\mu m^3$ in XYZ) per second for 80 volumes, for a total imaging time of ~2 min. **a** Maximum intensity projection views of the cell images at one-time point and overlay with representative trajectories of granule movement (orange lines). **b** Time series of the trajectories in **a. c, d** Histograms of estimated diffusion coefficients and velocities of all trajectories ($n = 185$) found in cell 1 and cell 2. **e–g** Dual-color volumetric imaging of live RBL-2H3 GFP-FasL cell, where the cell membrane is labeled with CellMask DeepRed and the granules contain GFP-FasL using an imaging speed of ~8.3 volumes/s for 80 volumes for a total time of 10 s. **e** Maximum intensity projection views of the cell images at one-time point and overlay with representative trajectories of granule movement (orange lines). **f** Time series of the trajectories in **e**. To indicate the current positions of the tracked particles, we used lighter orange lines represent past trajectories. **g** Histograms of estimated diffusion coefficients of all trajectories ($n = 48$) in the cell. **h** Cumulative probability of the estimated diffusion coefficients under normal (**a–d**) and stressed (**e–g**) imaging conditions. 400–500 trajectories with a diffusion coefficient $>0.001\,\mu m^2/s$ from four cells under each condition are selected.

But this increased distance breaks the 4f configuration between the two objectives (detection and remote objectives) which is critical to achieving aberration-free imaging. Future studies will investigate more compact designs that will better satisfy the 4f condition and will reduce the separation between the two foci at M3.

Following the footsteps of Hong et al.[45] we considered how our off-axis remote-focusing design might affect the collection efficiency[46,47] and therefore the achievable DL range (Supplementary Fig. 10). As it turns out, because in our current design the two images after the Obj2 are formed at the far end of the FOV, some of the rays always miss the Obj2 when we try to defocus. Our trigonometric analysis suggests that assuming a 90% collection efficiency as the acceptable limit, the scan range allowed should have been 253 μm (based on collection efficiency alone) (Supplementary Fig. 10c). However, we know that is not the case: as of now, our DL range is limited by the mismatched 4f which limits it to only 70 μm. Based on this analysis, it is imperative that to get the most out of the DL range and the collection efficiency of the remote-focusing system, while designing, the goal should be to not only maintain a strict 4f condition between L4 and L5 but also reduce the ΔL. As such we found that for a particular NA, having a lower magnification on the remote-focusing objective has two intertwined advantages: (1) it allows the user to choose a longer L5 which reduces the angular burden on the S and P paths (2) assuming a similar field number (FN), lower magnification objectives have larger FOV ( = FN/Magnification) which reduces the requirement of smaller ΔL to begin with. Our future designs will take these aspects into consideration and investigate newer designs that aim to reduce the angle between the S and P beams while adhering to the 4f condition between L4 and L5.

Of note, our approach offers several advantages over the existing axial refocusing methods. First, it provides an extended, aberration-free scan range for high numerical aperture (NA) optics. This is a significant benefit when compared to techniques based on deformable mirrors (DMs), where our method approximately doubles the axial scan range of DMs[48]. Second, it is wavelength-independent, which makes it suited for simultaneous multicolor imaging when compared to SLMs and tunable lenses. Additionally, unlike SLMs which depend on polarization, our arrangement is not dependent on the polarization of the fluorescence. Furthermore, unlike SLMs, which are typically slow (especially the nematic liquid crystal ones), and even their faster counterparts (ferroelectrics) tend to be less effective, our method allows for imaging speed that is only limited by the sCMOS's framerate.

Although recent advancements in oblique plane microscopy (OPM), which incorporates the benefits of LSMs to the convenience of single-objective microscopes, have achieved speeds comparable to our method, our technique presents several notable advantages. In OPM, the de-scanning of the returning fluorescent light leads to skewed images. Before these images can be viewed, they require intensive de-skewing processes[2,35,49–52]. On the other hand, our approach captures 3D volumes in a conventional orthogonal setup. This is achieved by recording high-speed images while sweeping the light sheet through the sample. Each frame captured by the camera represents an optical cross-section of the specimen. As a result, the 3D image stacks generated using our method are immediately available for viewing. They may benefit from an optional deconvolution, but there's no delay caused by necessary post-processing. Furthermore, many recent OPM setup employ a third objective which requires expensive objectives like 'Snouty' or 'King Snout'[2,35,49–51]. Our setup on the other hand does not have this requirement and our secondary objective performs the role of a tertiary objective. Moreover, while not demonstrated explicitly here, our method can be employed to achieve isotropic resolution, a feat the OPM cannot achieve.

Compared with the original Botcherby's remote focusing setup, our pmRF module folds the beam path between the detection and remote objectives. This configuration complicated the optical alignment. A potential solution is to arrange both objectives inline in a 4f configuration. Furthermore, we note that although an all-optical design has its merit of simplicity and robustness, using an objective lens in the pmRF module introduces ~30–40% light loss (Supplementary Fig. 5) compared with the axial scanning techniques based on DMs, future development of objective with high-transmission efficiency is desirable.

Finally, it is our firm belief that owing to its generalized design, we envision our method has the potential to transform many popular microscope modalities like confocal, 2-photon, and the rapidly emerging field of light sheet microscopy, by reinventing how they perform scanning in the axial dimension.

## Methods
### Optical setup
The illumination arm consists of two laser sources (Coherent Sapphire 488 nm and Obis LX 637 nm) which were combined with a dichroic beam splitter (LM01-503-25, Semrock). To clean up the beams, the beams were focused through a 50-μm pinhole (P50D, Thorlabs) by a 45-mm achromatic doublet (AC254-045-A, Thorlabs) and then recollimated using a 150-mm achromatic doublet (AC254-150-A-ML, Thorlabs). The original beams were expanded by 9 folds with a 3× Galilean beam expander (GBE03-A) before being focused with a cylindrical lens (ACY254-50-A, Thorlabs), onto a resonant mirror galvanometer (CRS 4 kHz, Cambridge Technology), driven by a 12-volt power supply (A12MT400, Acopian), to wobble the light sheet. The one-dimensional focus was then recollimated with a 100-mm achromatic doublet (AC254-100-A-ML, Thorlabs) and hit the galvanometric scan mirror (GSM) (GVS111, Thorlabs), driven by a 15-volt power supply (GPS011, Thorlabs), for rapid shifting of the light sheet along the detection arm. This galvanometric mirror was conjugated to the back pupil of the objective lens (Nikon 40x/0.8 NA) through 100-mm and 200-mm achromatic doublet (AC508-100-A-ML and AC508-200-A-ML, Thorlabs).

In the detection arm, the same objective lens (Nikon 40x/0.8 NA) in an orthogonal setup was used and pupil-matched to the scanning objective lens (Nikon Plan Apo 20x/0.75 NA) through a 200-mm tube lens (L4) (TTL200-A, Thorlabs) and a 300-mm achromatic doublet (AC508-300-A-ML, Thorlabs). A 50:50 polarizing beam splitter (PBS) (10FC16PB.3, Newport), splits the beam in S and P polarized light. Using mirrors M1 and M2 these light paths were then launched at an angle towards the Obj2 (Supplementary Fig. 2). It is critical to minimize the angle of the launch. Both experiment and simulation predicted that we used 8 degrees as the launch angle (called θ) (Fig. 2a, inset of Supplementary Fig. 2). The S and P polarized light passed through a quarter waveplate (AQWP10M, Thorlabs) and were focused onto a mirror positioned at the focus of the scanning objective lens. The mirror (PF03-03-P01 - Ø7.0 mm Protected Silver Mirror, Thorlabs) was attached to a voice coil with a travel of 10 mm, positional repeatability of fewer than 50 nanometers, and a response time of fewer than 3 milliseconds (LFA-2010, Equipment Solutions). Then the reflected light was recaptured by the same scanning objective lens and quarter-wave plate to rotate the beam's polarization state. Afterward, the light was directed toward an sCMOS camera (Hamamatsu Orca-fusion BT) by reflecting from the same cube polarizing beam splitter and a 300-mm achromatic doublet (AC508-300-A-ML, Thorlabs). For emission filters, we used two long-pass filters (FF01-525/30-25, and BLP01-647R-25, Semrock), for blue, and far-red, respectively. To image dual channels simultaneously, the FOV was separated into half using dichroic mirrors (DMLP605R, Thorlabs) between the 300-mm achromatic doublet and the camera. We immersed the specimen and the illumination and detection objectives in a chamber designed using Adobe Inventor and machined through Protolabs (R). The LFA and GSM, in the detection and illumination arms respectively, were synchronized together to always keep the translated light sheet in the focus of the detection objective lens to acquire a 3D stack of the specimen.

## Overlaying S and P images with sub-pixel accuracy

Two identical images – one corresponding to S and another to P polarized light – are formed at the camera and added incoherently to generate the final image. We used a custom-written MATLAB script to monitor the offset between the two images in near real-time while adjusting the positions of M1 and M2 (Supplementary Fig. 6).

The offset between the two images using a cross-correlation-based algorithm as was used in Wester, M.J. et al.[53], which achieves sub-pixel accuracy by fitting second-order polynomials through the peak of the scaled cross-correlation between the S and P polarized images.

Initially, an image is acquired as a reference by obstructing one optical path (either S or P). Subsequently, the alternative optical path is used to collect new images. The shift between each new image and the reference image is then measured using the method described in Wester, M.J. et al.[53] and is available as the MIC_Reg3DTrans.findStack-Offset method in the MATLAB-instrument-control toolbox[54]. While new images are being collected, mirrors M1 and M2 are adjusted to minimize the shift.

## Microscope control

A Dell Precision 7920 computer with two processors Intel(R) Xenon(R) Silver 4210 R CPU having a processing speed of 2.40 GHz and 2.39 GHz and was integrated with 128 GB RAM was used to acquire the microscope's data. An NVIDIA Quadro RTX 4000 Graphics processing unit (GPU) with dedicated memory of 8 GB and shared memory of 63.8 GB (GPU memory of 71.8 GB) was also integrated into the system. 64-bit operating system ×64-based processor facilities the system to operate. LabView 2020 64-bit allowed us to work with the required software, including the LabView Run-Time Engine, Vision Run-Time Module, Vision Development Module, and other required drivers like NI-RIO drivers (National Instruments). DCAM-API software was used for the Active Silicon Firebird frame-grabber to actively interfere with the scientific complementary metal-oxide semiconductor (sCMOS) camera (ORCA-Fusion BT Digital CMOS camera, model: C15440-20UP) manufactured by Hamamatsu, Japan. It generated deterministic transistor logic (TTL) trigger sequences through 150 Watts shutter instrument (100–240 V ~ 50/60 Hz; model: MP-285A) with a field programmable gate array (FPGA) (PCIe 7852R, National Instruments). The generated triggers controlled the resonant mirror galvanometers, placement of the stage, voice coils, blanking and modulation of laser, firing camera, and other external triggers. K-Hyper Terminal software facilitated engaging LFA with the system hardware. Some key features along with some routines under the agreement of material transfer were licensed by the Howard Hughes Medical Institute's Janelia Farms Research Campus.

## Sample preparation

Bead sample:200 nm beads embedded in 2% agarose gel were used for microscope resolution assessment. To make 2% agarose gel, 2 g of agarose powder (A9045-25G, SIGMA life science) was mixed with 100 mL water and swirled thoroughly before putting into the microwave oven to heat. Once the solution boiled and got completely clear and the agarose was dissolved, we should remove the solution from the oven and let it cool down. Then, 200 nm beads (YG, Polysciences) were mixed with water with a ratio of 1/100 to form a solution of the 200 nm beads. It was sonicated before mixing with the molten 2% agarose gel with a volumetric ratio of 1/10. Then this molten combination was poured into the cubic mold where the sample holder was placed and sat there for a few minutes to dry and form a 1cm³ cubic sample (200 nm beads embedded into 2% agarose gel) attached to the sample holder.

Cell samples: RBL-2H3 GFP-FasL cells were cultured in Gibco Minimum Essential Media (MEM) media supplemented with 10% heat-inactivated Fetal Bovine Serum (FBS), 1% Penicillin/Streptomycin, and 1% L-glutamine[37]. The cells were primed with 1 μg/ml anti-DNP-IgE[55] (Fig. 3e–g) or anti-DNP-IgE-CF640R (Fig. 3a–d) and seeded at a density

of 100,000 cells per well in 12 well dish over 5 mm glass coverslips and incubated with 5% $CO_2$ at 37°C overnight. IgE-CF640R was prepared using CF640R NHS-ester (Biotium #92108). For fast imaging experiments, the cellular membranes of anti-DNP-IgE primed cells were labeled with CellMask™ Deep Red Plasma Membrane Stain (Thermo Fisher Scientific #C10046, 5 mg/ml, 1000X) according to manufacturer's instruction for 10 min in modified Hank's balanced salt solution (HBSS) (additional 10 mM Hepes, 0.05% w/v BSA, 5.45 mM glucose, 0.88 mM MgSO4, 1.79 mM CaCl2, 16.67 mM NaHCO3) and rinsed with HBSS. Cells were stimulated with 1 μg/ml DNP-BSA in the sample chamber. Data was acquired in 2-min captures for up to 15 min post antigen treatment.

## Sample mounting

Cell samples on 5 mm coverslips were loaded onto the holder as depicted in Supplementary Fig. 7. In the sample holder, two metal wires were designed to clamp the coverslip tightly. This sample holder was attached to the *XYZ* Translation Stage with Standard Micrometers using a rotation mount. As a result, the coverslip had four degrees of freedom, including the translation on the *X-Y-Z* axis to locate the cells while imaging and the rotation around the X-axis to face the coverslip with the desired angle relative to the illumination and detection objectives. Here, the coverslips were faced 8 degrees relative to the optic axis of the detection objective (Supplementary Fig. 8). In order to minimize the buffer volume for live cell imaging, the 6 ml chamber was designed to immerse the sample, illumination, and detection of objective lenses into it (Supplementary Fig. 7).

## Image processing pipeline

Data were analyzed with the custom script written in MATLAB. The procedure for quantifying the microscope's resolution from fluorescence bead data is as follows: (1) A 3D-PSF model was generated from the raw data. PSF model was generated using the voxel-based PSF modeling method from uiPSF[56]. In this method, a 3D matrix representing the PSF model was extracted from multiple beads stacks using inverse modeling. For generating the light-sheet PSF model, a beads scan was collected by imaging beads in agarose gel at axial positions from −40 μm to 40 μm with a step size of 500 nm. Then beads within 5 μm around the light-sheet waist and 40 μm within the center of the scan range were selected for generating the PSF model. (2) The light-sheet region was cropped from each slice of the raw data. The FOV (length) of the light-sheet region was defined by the distance to the waist of the light-sheet where the axial resolution increases by 2 times. We set the light-sheet FOV to be ~8 μm. Note that the light-sheet region translated along its width direction (the *Y*-axis) while it was being scanned in the axial direction (*Z*-axis relative to the detection objective) (Supplementary Fig. 8). Therefore, the light-sheet region to be cropped was also shifted in *Y* accordingly (Supplementary Video 1). This is done to ensure that the detection light paths are not aberrated or hindered by the glass coverslip we tilted the coverslip slightly (by 8 degrees) while still trying to maximize the light-sheet confocal parameter and resolution as shown in Supplementary Fig. 8. Therefore, we traversed the light-sheet waist to follow along the surface of the coverslip. This then required us to break the conjugation of the galvo and the illumination objective ever so slightly that we could get the light-sheet scanning volume orthogonal to the coverslip. (3) the cropped was divided into segments with an axial dimension of 5 μm. For each segment, candidate beads were selected and their FWHMs along each dimension were estimated from Gaussian fitting of their intensity profiles along that dimension. 4) the measured FWHMs were used to quantify the resolution of the microscope as shown in Fig. 2d.

Dual-color live-cell data was processed as follows: (1) cell signal from each color channel was cropped with a user-selected region. (2) for each color channel, the *XYZ* drifts of the data stack at each time point relative to the reference data stack were estimated, where the

maximum-intensity projection (MIP) along each dimension of the two data stacks was generated and the 2D shift between each pair of the MIP images was calculated through cross-correlation. (3) an average of the *XYZ* shifts from both channels was used to correct the drift between time points. (4) the *XYZ* shift between the two-color channels was calculated by first averaging over the time dimension for each color channel, then estimating the shift from the MIP images as in step 3. Then register the two channels by applying the estimated shift. 5) after drift correction and channel registration, the resulting image stacks were deconvolved with the 3D-PSF model generated from the bead data using Richard-Lucy deconvolution from MATLAB. 6) the resulting image stacks were interpolated along the scan dimension to ensure equal pixel size along all three dimensions. 7) to reduce noise and correct photo-bleaching, the deconvolved images were subtracted by a background value with negative pixel values set to zero and divided by a normalization factor equal to the 99.95 quantiles of all pixel values in the corresponding time points and color channel.

## Quantification of Light-sheet dimension

To quantify the light-sheet dimension, bead data in agarose gel were collected while varying the widths of the physical slit in the illumination light path. This slit modulates the numerical aperture (NA) of the illumination. At each slit width, we estimated the FWHMs in *XYZ* for all selected beads as described above, however, here we used the full FOV of the color channel for bead imaging. As the position of the light-sheet waist shifted in *Y* with respect to the axial dimension, we corrected the *Y* coordinates of the selected bead by $y'_{cor} = y_{cor} - a z_{cor}$, where $a$ is the *Y* shift by moving one pixel in *Z* (Supplementary Fig. 8). Then we fitted the $FWHM_z$ verse $y'_{cor}$ for all selected beads with a polynomial function (Supplementary Fig. 4). The FOV (length) of the light-sheet was found when the $FWHM_z$ was twice the minimum from the polynomial fit.

## Magnification calibration

One brightfield image of the calibration target was captured at each of the galvo positions from -40 μm to 40 μm with a step size of 10 μm. The target image consisted of parallel line segments, we cropped a region of 700×700 pixels from each image (Supplementary Fig. 9a, b). We then calculated the affine transformation (from the Dipimage toolbox)[57] of each image with respect to a reference image. The zoom factors from affine transformation were used to quantify the relative magnification between each image to the reference image. The absolute magnification of one image was calculated as follows: crop a narrow section of multiple parallel lines, obtain the intensity profile by averaging over the line dimension, smooth the intensity profile by applying a running average with a window size of 30 pixels, find all peaks from the smoothed intensity profile (Supplementary Fig. 9c), calculate the average distance (in pixels, denoted as $\Delta d$) between consecutive peaks, as the distance between consecutive parallel lines is 10 μm, then the pixel size at the sample plane can be estimated from $10/\Delta d$ μm, therefore the magnification can be calculated from the pixel size of the camera divided by pixel size at the sample plane.

## Ray tracing

Ray tracing was based on geometric optics with paraxial approximation. The ray propagation was calculated using the ABCD matrices. Two matrices were used, the translation matrix,

$$M_d = \begin{bmatrix} 1 & 0 \\ d/n & 1 \end{bmatrix} \tag{1}$$

and the matrix of a thin lens,

$$M_f = \begin{bmatrix} 1 & -n/f \\ 0 & 1 \end{bmatrix} \tag{2}$$

where $d$ is the translation distance, $f$ is the focal length of the thin lens and $n$ is the refractive index of the propagation medium. For our system, $n$ is 1.33 before the detection objective (including the objective) and $n$ equals 1 for the rest of the ray tracing. The starting point of each ray was represented by a vector of $[n\alpha, y]^T$, where $\alpha$ and $y$ are the angle and the y position of the ray with respect to the optical axis. The propagation of the ray is then calculated from

$$\begin{bmatrix} n\alpha' \\ y' \end{bmatrix} = M \begin{bmatrix} n\alpha \\ y \end{bmatrix} \tag{3}$$

For a defined FOV, we selected three field points, two mark the edge of the FOV and one at the optical axis. For each field point, we generated three rays at different angles that will intersect three points at the pupil plane, where two points mark the edge of the pupil and one at the center of the pupil. Rays from the same field point were colored the same. The optical axis after the polarizing beam splitter was rotated by 45 degree to be along the splitting plane of the PBS. The ray tracing after the PBS was done by first transforming the ray coordinates to the ones defined by the optical axis and propagating the ray with the ABCD matrix, then transforming it back to the global coordinates. Except for the distance between the tube lens (L4) and the detection objective (Obj1) (denoted as $d_1$), the rest distances between consecutive optical elements were measured with a ruler. The angle between the chief rays of the S and P-polarization ($\theta$ in Fig. 2a) was set when the input and output beam diameters at the remote focusing objective were minimal. The distance $d_1$ was set when the relative magnifications from ray tracing matched with the measured ones (Fig. 2b). The central position of the scan range, the distance of the objective to the detection objective (denoted as $S_1$), was set when the absolute magnification from ray tracing matches with measured one. Here $S_1 = 6.695$ mm, which was 45 μm away from the designed focal plane of the detection objective.

## Reporting summary

Further information on research design is available in the Nature Portfolio Reporting Summary linked to this article.

## Data availability

The data presented in this paper are publicly available on Zenodo at https://zenodo.org/records/10864759.

## Code availability

The MATLAB scripts used in this paper are available at https://github.com/ChakraOpticsLab/RemoteFocusing.

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

## Acknowledgements
This work was supported by the University of New Mexico (Start-up Grant) (T.C.), NIH R35GM151152 (T.C.), NIH P30CA118100 (T.C. and K.L.), and NIH R35GM126934 (D.S.L.). We thank Derek Rinaldi for generating the IgE-CF640R. This work was conducted with support from the University of New Mexico Office of the Vice President for Research Program for Enhancing Research Capacity, was supported by grants from NVIDIA, and utilized an NVIDIA A6000 GPU.

## Author contributions
T.C. conceived the idea of lossless remote focusing in the detection arm. H.D. and T.C. designed and built the remote focusing unit. H.D. and T.C. designed, built, and operated the microscope. H.D. and Sh.Li performed image analysis. Sh. Li and J.P. have theoretically demonstrated that achieving 100% conversion from unpolarized to polarized light is not feasible. D.J.S. and K.A.L. provided the MATLAB code for the fine alignment. D.S.L., Sh.Lu., and R.M.G. prepared RBL cells for imaging. H.D. imaged RBL cells labeled with DeepRed CellMask. H.D. and A.K.N.Sh imaged the RBL cells labeled with IgE-CF640R. H.D., Sh.Li., and T.C. wrote the manuscript. All authors read and provided feedback on the final manuscript.

## Competing interests
T.C. and H.D. have filed a patent application (United States Patent and Trademark Office application number 63/397,714) for the remote focusing setup mentioned here. The remaining authors declare no competing interests.
