## [Peer Review File · Nature Communications]

Axial de-scanning using remote focusing in the detection arm of light-sheet microscopyEditorial Note: Parts of this Peer Review File have been redacted as indicated to maintain the confidentiality of unpublished data.

Reviewers' comments:

Reviewer #1 (Remarks to the Author):

Dibaji et al. describe an optical method to incorporate remote focusing for Z-scanning into the collection arm of a fluorescence microscope. Standard remote focusing architectures rely on polarization to direct light to and from the remote focusing optics, meaning they would incur a 50% light loss in the (unpolarized) collection arm of a fluorescence microscope. The method described here is unique in that there is no polarization-induced light loss. Compared to mechanical Z-scanning methods (e.g., translating the objective lens or sample), this method allows for faster scanning and does not require translating the sample. Compared to optical methods (e.g., SLM / DM) this method can achieve larger scan ranges and reduced aberrations. The authors implement their approach in a light-sheet microscope, characterizing the optical performance (PSF) of the device and demonstrating live cell imaging.

I find this to be a very elegant technique. Through a clever arrangement of only two additional mirrors, the setup can recover all the light that would ordinarily be lost at the PBS in a conventional remote focusing setup. I agree that this technique should have broad applicability in a number of microscopy techniques and will be of interest to readers. The manuscript is technically sound, and the concept is novel to my knowledge. However, I find various parts of the manuscript confusing and believe the clarity needs to be improved prior to publication. Several suggestions are below:

I am confused by the 8 deg angle mentioned in multiple places. I assume this is intentional (as opposed to a misalignment) – is this a geometric constraint needed to maintain an orthogonal light-sheet geometry while imaging cells on a coverslip? Similarly, why / how is the sheet translated in the y-direction during z-scanning? Can the authors explain this? Perhaps a diagram of the collection and illumination beams relative to the sample and coverslip would be helpful.

Characterization of the optical system: The authors state that bead measurements were taken from “MIP of 10 slices, each separated 500 nm”. I am confused by the axial spacing of 500 nm – is the axial dimension undersampled (given 654 nm axial resolution)? The YZ images shown seem to show square pixels spaced by ~167 nm as opposed to 500 nm. Can the author clarify?

Image processing pipeline: The authors state “A 3D-PSF model was generated from the raw data” – please elaborate, it’s not clear to me how the 3D model was generated.

Quantification of Light-sheet dimensions: the authors state that bead data was collected “at different slit widths” – it’s not clear what this means.

Magnification calibration: “...as the distance between consecutive parallel lines is 10 mm, then the pixel size at the sample plane can be estimated from $10/\Delta dd$ mm, therefore the magnification can be calculated from the pixel size of the camera divided by pixel size at the sample plane.” – should this be μm instead of mm in both cases?

Fig. 1b: Some labels are missing (e.g., tube lens, camera), and the naming of some components is inconsistent between the figure and various sections of the text (Obj2/RF objective/pmRF objective). This made things harder to follow.

Fig. 1e: Please add more details: What do the errors represent (standard deviation, etc.)? Are bead images and curves single beads or many beads averaged together? The red/blue annotations and the apparent curve widths don’t seem to completely match. For instance, annotations on the right-most plot indicate the red curve is ~66% wider, but the two curves look almost the same size. Units are missing on the plot axes.

Fig. 2d: Also missing some details: What is the meaning of the dotted line vs. the circles vs. the error bars? How many beads are shown at each Z position, and how were these chosen?

Fig. S2: The inset includes “ $d = 300$ (f5)”. What does ‘f5’ mean? It looks like this is the 4f case?

Fig. S2: There is an arrow with the letter ‘a’ – what does this mean?

Writing:

There are a number of typos and grammatical errors throughout the manuscript that made it challenging to follow in some places. A few are noted below, but the paper should be copy edited carefully prior to publication.

“To avoid the slow translation of bulky objectives or the sample stages, several attempts, employing variable-focus (vari-focus) lenses, mechanical mirrors, and acousto-optics modulators have been proposed to refocus the light for 3D imaging.” Missing comma after ‘modulators’

“To the best of our knowledge, because, using purely linear optical elements, lossless conversion of unpolarized light into a single polarized state is not yet possible (Supplementary Note 1), manipulating the fluorescent light using the optical isolators is unfeasible.” – consider rewording, hard to follow.

“Fig. 2c shows the maximum intensity projection (MIP) of beads (from 10 axial slices, each slice spaced 500 nm) separated by 30 μm for S, P, and S+P across the scan range, after 10 iterations of Richardson-Lucy (RL) deconvolution.” – consider rewording this, it took me a long time to understand what was separated by 30 μm (groups of beads)

“...both the S and P-polarized images rendered onto the camera exhibit identical FWHM” – the term ‘identical’ is confusing given that the numbers for S & P are different. Perhaps ‘comparable’, etc. would be better? Similarly in the next paragraph: “...thereby resulting in an identical ‘S+P’ image”

Reviewer #2 (Remarks to the Author):

Paragraph beginning: “Unlike the adaptive optics or DM-based approaches...”

In this paragraph you need to address the unfolded geometry discussed by Botcherby in his original paper. You may decide to discuss the temporal limitations of the unfolded geometry (moving the entire objective mass) but you must make mention of the availability of a remote focusing configuration that can collect 100% of the light. Equally it is inaccurate to state that microscopes using pmRF carrier axial scanning necessarily lose 50% of the fluorescence light. Please change this.

Paragraph beginning: “Thirdly, there exists an inverse relationship between...” this paragraph states the trade-off between adhering to a 4F system or minimising the angular separation of the two polarisation images. Please state explicitly which side of this trade-off the current design adheres to.

“Characterization of the optical system”

Please state why it is valid to implement the paraxial approximation for hi na lenses. The paraxial approximation is only valid and the small angle limit ($\sin(\theta) \sim \theta$). Please calculate the size of errors you would anticipate and applying the paraxial approximation to the large acceptance angles of the lenses used in this experiment.

Whilst the specifics of the detection and remote objectives are given in the methods section they should first be specified in this section of the paper (including the NA and model) to enable the reader to make an informed comparison between the results of the simulation with that expected from the experiment

Figure 1: Typo: 'exist' = 'exit'

As we now know, the resolution of a remote focusing system varies across the field of view. Please state explicitly where the beads analysed in the performance of your system (Figure 2) were located within the field of view of your detection objective.

Figure 2 caption: "Z direction exhibits less resolution in the axial direction controlled by the light sheet waist". There seems to be a misunderstanding here. The axial extent of the fluorescent bead image is determined only by the dimensions of the bead and not the light sheet. The axial resolution would only be limited by the light sheet thickness in the case of an extended object. Please correct this part of the text.

Figure 2: "The minimum lateral resolution, 394 nm, occurs at the center of the scan range". This statement is true for the Y direction but not true for the X direction where the minimum is closer to 15 microns in Z. Please provide a justification for why this might be.

Figure 2b. This figure shows the Z location in terms of the galvo position. As the reader has no way of mapping between the 'galvo' position (light sheet) to the 'Z' position (mirror) please redraw Fig 2B in terms of the Z units for consistency.

Deconvolution: Deconvolution algorithms have no place in this work. Richardson Lucy deconvolution actively de-blurs the raw bead images, changing completely what it is that you are reporting. Please redraw Figure 2d for the raw bead image data to provide a true reflection of the system performance.

"In the axial direction (the YZ view) the PSFs are limited by the Gaussian light sheet's waist". Please see earlier comment.

Discussion

Paragraph beginning: "Although recent advancements in single-objective oblique plane microscopy (OPM) have achieved speeds comparable to our method". It is important to acknowledge the advantages of a single objective like sheet methods compared to orthogonal light sheet methods in

terms of sample accessibility. Many imaging experiments in biomedical science are not possible using the orthogonal objective like sheet method.

In the new configuration it is not possible to inject the excitation light via the polarising beam splitter. Please comment on how the new detection configuration would integrate other optical sectioning technologies; specifically spinning disc, single objective light sheet, 2 photon and structured illumination.

Please comment on the feasibility of injecting excitation light into the 'detection' objective via a dichroic mirror to enable single objective light sheet illumination and polarisation insensitive fluorescence detection.

Methods: Optical setup: It is not clear to me why you have chosen objectives with different magnifications. In order for the remote focusing principles to apply, the only magnification of the stigmatic remote image is due to the index difference between the water sample and the air immersion mirror (i.e. a magnification of $4/3$). By using a 40X imaging objective coupled to a 20X remote objective you introduce an extra factor of 2 into the magnification.

References: many of the references are poorly formatted some with author names missing completely. Please reformat reference list.

Reviewer #3 (Remarks to the Author):

In the manuscript "Axial de-scanning using remote refocusing in the detection arm of light-sheet fluorescence microscopy", Dibaji et al. present a novel optical configuration for minimizing light loss in a folded aberration-free remote refocusing relay. In the presented design, the collected fluorescence, split into s- and p-polarizations by a polarized beam splitter, is reflected into the pupil of the second objective in the remote-refocusing relay. The light is back-reflected along the same path using a mirror perpendicular the optical axis of Obj2, reversing the polarization and then coupled out of the remote refocusing relay by the PBS, recombined at the detector. This approach allows the use of light with the polarization that is typically lost in a folded remote refocusing relay, improving the optical collection efficiency. The method proposed in the manuscript is a clever and original solution to the main trade-off when using folded remote-refocusing in the detection path of

a fluorescence microscope, and although the implementation involves its own trade-off, it may be a useful solution for certain applications.

The manuscript presents a detailed characterization of the system design and performance, as well as some proof-of-concept applications, and discussion of the trade-offs. The manuscript is well-written with good quality data presentation and figures, and would be of interest to a wide audience within the microscopy development community. I believe the manuscript would be suitable for publication in this journal, with some minor revisions.

Main points:

- On the second page, the authors state that “in the detection arm, the fluorescence is unpolarized in nature”. This is not always entirely true – for fluorophores with high levels of fluorescence anisotropy and polarized excitation light, the fluorescence can be partially polarized, with less than 50% loss during PBS double pass. The manuscript would benefit from adding a brief discussion of the role of fluorescence anisotropy on light loss through the polarized beam splitter when remote-refocusing in the detection path. In this case transmission can be maximized using a HWP before the PBS (see Sparks et al (2020), DOI: 10.1002/jbio.201960239).

- The authors state that in order to minimize the loss of FOV in the proposed configuration, the design benefits from a small angle θ , as this minimizes the separation of the two remote images formed by Obj2, enabling the use of a smaller mirror for refocusing and reducing the loss of FOV.

- o I would rephrase the last sentence which states a “smaller ΔL ...guarantees that both images fit within Obj2 FOV” – as this does not guarantee this. Even for a very small L , Obj2 can limit overall FOV, depending on objective choice.

- o The authors propose breaking the 4f system arrangement to increase the distance between Obj2 and the PBS, reducing θ , and that the compromise in the current design still achieves diffraction-limited (DL) resolution. However, it is not clear from the manuscript as to what the DL remote refocusing range would be when maintaining the pupil-matched 4f remote refocusing configuration as opposed to minimizing θ . As the DL range is reduced, image contrast away from the nominal focal plane is also reduced – which may counteract the benefit of improving the collection efficiency in folded remote refocusing in detection using the configuration proposed by the authors.

- o The manuscript would benefit from including a more detailed discussion of the trade-off between FOV, collection efficiency through the PBS, and DL range of the remote refocusing at various NA, as this would be the key consideration when deciding whether the proposed configuration would be useful. However, I don't think that a quantitative analysis is necessarily required within the scope of the manuscript which is more focused on a proof-of-concept demonstration of the principle.

- o Using the deconvolved bead volumes to evaluate the DL range can be misleading as the quoted axial resolution of deconvolved beads may be subject to the deconvolution parameters and not fully representative of the raw optical performance of the system. I would propose using the raw bead data, and accounting for bead size and pixels size to estimate the PSF FWHM.

Minor comments:

- Page 2, Line 2 – The authors state that “for an objective with an NA of 0.8, the maximum axial scan range that DM based technique can generate is approximately 40 μm ”. However, in work by Wright et al (2021), (DOI: 10.1088/2515-7647/ac29a2), a 100 μm scan range is achieved with a 0.8 NA lens (with a Strehl ratio of >0.6 over 77 micron refocus range). A similar scan range is achieved in a LSM implementation with a slightly lower NA by Hong et al (2023) (DOI: 10.1063/5.0125946)
- Results, second paragraph, Line 2 “M3 consistently moves in parallel with the focal plane of Obj2” – I think it might be more clear to state that M3 translates along the optical axis of O2, or perpendicular to the nominal focal plane of O2.
- Last line of page 3 (“The small deviation is limited by the geometry...”). Referring to a magnification mismatch of 0.07 (5%) as “small” is rather subjective for a system with the current NA. Mohanan and Corbett (2020, 2023) and Hong et al (2023) present a detailed characterization of remote refocusing performance and impact of magnification mismatch on the diffraction-limited refocusing range at various NA systems, suggesting that even a 1% mismatch can reduce the DL range to half its maximum value.
- In Supplementary Figure 9, describing magnification calibration, there appears to be some minor clock-wise rotation of the calibration target relative to the FOV x-y orientation. Is this accounted for when estimating the peak separation of the features?
- 4th paragraph in Discussion: “ The OPM setup necessitates a third objective...” – this is not strictly true – OPM can also be implemented using folded remote refocusing with a tilted mirror in remote refocus space – see the dOPM setup by Sparks et al (2020) (DOI: 10.1364/BOE.409781)

Grammar/formatting:

- Line 2 of main text – “BioPhotonics” – no need for capitalization at start or middle of word
- Two sentences on page 2 - not sure whether the use of italics is deliberate, but is perhaps not necessary
- Line 6 of discussion “Carryout” – 2 words
- Line 3 of Results section - “Botceherby” – Botcherby

Reviewers' comments:

Reviewer #1 (Remarks to the Author):

Dibaji et al. describe an optical method to incorporate remote focusing for Z-scanning into the collection arm of a fluorescence microscope. Standard remote focusing architectures rely on polarization to direct light to and from the remote focusing optics, meaning they would incur a 50% light loss in the (unpolarized) collection arm of a fluorescence microscope. The method described here is unique in that there is no polarization-induced light loss. Compared to mechanical Z-scanning methods (e.g., translating the objective lens or sample), this method allows for faster scanning and does not require translating the sample. Compared to optical methods (e.g., SLM / DM) this method can achieve larger scan ranges and reduced aberrations. The authors implement their approach in a light-sheet microscope, characterizing the optical performance (PSF) of the device and demonstrating live cell imaging.

I find this to be a very elegant technique. Through a clever arrangement of only two additional mirrors, the setup can recover all the light that would ordinarily be lost at the PBS in a conventional remote focusing setup. I agree that this technique should have broad applicability in a number of microscopy techniques and will be of interest to readers. We sincerely appreciate this comment and believe that this method can truly change how 3D volumetric imaging is done across many microscopy techniques. We sincerely thank the reviewer for commending the importance of our work. The manuscript is technically sound, and the concept is novel to my knowledge. However, I find various parts of the manuscript confusing and believe the clarity needs to be improved prior to publication. Several suggestions are below:

I am confused by the 8 deg angle mentioned in multiple places. I assume this is intentional (as opposed to a misalignment) – is this a geometric constraint needed to maintain an orthogonal light-sheet geometry while imaging cells on a coverslip? Similarly, why / how is the sheet translated in the y-direction during z-scanning? Can the authors explain this? Perhaps a diagram of the collection and illumination beams relative to the sample and coverslip would be helpful.

We agree with the reviewer that the “8-degree” information is confusing and sincerely apologize for not clarifying that in our manuscript. As it turns out we have two separate but independent places where 8-degree angles come into play: (1) the angle between the S and the P beam before the Obj2, and (2) the angle between the lightsheet translation and the Z-axis.

While the 8-degree angle between the S and P beam was a design requirement, the reviewer is correct in his/her assumption that the 8-degree angle between lightsheet translation and the Z-axis was a geometric constraint needed to maintain an orthogonal lightsheet geometry. This is because, to ensure that the detection light paths are not aberrated or hindered by the glass coverslip we tilted the coverslip slightly (by 8 degrees) while still trying to maximize the volume spanned by the lightsheet waist, as shown in Sup. Fig. 8. Therefore, we traversed the lightsheet waist to follow along the surface of the coverslip. This then required us to break the conjugation of the galvo and the illumination objective ever so slightly that we could get the lightsheet scanning volume orthogonal to the coverslip. As suggested, we have now clearly depicted this in our Supplementary Figure 8. [Line 465-469]

Supplemental Figure 8: YZ view of MIP of stacks of 200 nm beads. a) The light sheet translates in Y while scanned in Z direction. The angle between the light sheet translation and the Z-axis is 8 degrees. The beads are RL deconvolved and only the beads in the light sheet waist region are cropped and shown here. b) The orientation of the coverslip relative to the detection and illumination objectives was 8 degrees.

Characterization of the optical system: The authors state that bead measurements were taken from “MIP of 10 slices, each separated 500 nm”. I am confused by the axial spacing of 500 nm – is the axial dimension undersampled (given 654 nm axial resolution)? The YZ images shown seem to show square pixels spaced by ~167 nm as opposed to 500 nm. Can the author clarify?

We can understand why the Reviewer is confused: this is because we report the deconvolved numbers as our resolution. However, the raw images suggest that the waist of the lightsheet is ~900nm. Therefore, we chose 500 nm as our step size, although based on Nyquist sampling we probably should have chosen 450 nm. So, in that sense, the reviewer is correct the axial dimension is slightly under-sampled here.

Regarding the square pixels: To make the pixels isotropic we originally up-sampled these images in the z direction by 3.08 times using bicubic interpolation. However, understandably it was confusing, so now we show the raw unprocessed images with rectangular pixels.

Image processing pipeline: The authors state “A 3D-PSF model was generated from the raw data” – please elaborate, it’s not clear to me how the 3D model was generated.

PSF model was generated using the voxel-based PSF modeling method from uiPSF (<https://doi.org/10.1101/2023.10.26.564064>), In this method, a 3D matrix representing the PSF model was extracted from multiple beads stacks using inverse modeling. For generating the lightsheet PSF model, a bead-scan was collected by imaging beads in agarose gel at axial positions from -40 μm to 40 μm with a step size of 0.5 μm . Then beads within 5 μm around the lightsheet waist and 40 μm within the center of the scan range were selected for generating the PSF model. [Lines # 455-460]

Quantification of Light-sheet dimensions: the authors state that bead data was collected “at different slit widths” – it’s not clear what this means.

In our light-sheet microscopy, adjusting the slit width controls the numerical aperture (NA) of the illumination objective, affecting the Rayleigh range and width of the light sheet. A tighter slit

increases the FOV (Rayleigh range) but decreases Z-resolution (fatter light sheet). Here, we used the slit to have a larger FOV for imaging.

Magnification calibration: "...as the distance between consecutive parallel lines is 10 mm, then the pixel size at the sample plane can be estimated from $10/\Delta d$ mm, therefore the magnification can be calculated from the pixel size of the camera divided by pixel size at the sample plane." – should this be μm instead of mm in both cases?

We sincerely apologize for this TYPO. It is fixed now. [Lines 491-493]

Fig. 1b: Some labels are missing (e.g., tube lens, camera), and the naming of some components is inconsistent between the figure and various sections of the text (Obj2/RF objective/pmRF objective). This made things harder to follow.

The figure labeling (Fig. 1b) is fixed. pmRF objective/RF objective is changed to obj 2. [Lines # 161-167]

Fig. 1e: Please add more details: What do the errors represent (standard deviation, etc.)? Are bead images and curves single beads or many beads averaged together? The red/blue annotations and the apparent curve widths don't seem to completely match. For instance, annotations on the right-most plot indicate the red curve is ~66% wider, but the two curves look almost the same size. Units are missing on the plot axes.

We thank the reviewer for this comment. The errors represented in the plots were standard deviations, indicating the variability or spread of resolution measurements across multiple beads (20 beads). The curves represent the single bead but the numbers on the curves are an average taken from multiple beads to provide a more reliable statistical assessment of our system's performance rather than relying on single bead measurements, which could be more susceptible to anomalies.

There is no discrepancy in the apparent widths of the red and blue curves, if we consider the plus and minus standard-deviation to average values.

However, we feel that this method was confusing so now we just depict plots corresponding to one bead. In the main text we report numbers with statistical significance. [Lines#198-200]

Units were added to the plot.

Fig. 2d: Also missing some details: What is the meaning of the dotted line vs. the circles vs. the error bars? How many beads are shown at each Z position, and how were these chosen?

The circles indicate the mean resolution measurements obtained from the beads at each Z position, providing a point of reference for the average system performance at different depths. The error bars show the standard deviation from the mean resolution at each Z position, quantifying the spread of measurements and thus the consistency of the system's performance across the field of view. The dotted line represents the polynomial curve fitting over average data (circles). It serves as a trend to show resolution changes over the scanning range. [Line#219-224]

20 randomly selected beads were chosen for this analysis. The selection criteria for the beads at each Z position include ensuring that they are well-separated to prevent overlap in measurements

and sufficiently bright for accurate resolution assessment. It is performed by MATLAB code, where beads were chosen by cross-correlation between the maximum intensity projection of the PSF model along z and beads in each of 10 slices. [Line# 219]

Fig. S2: The inset includes “ $d = 300$ (f_5)”. What does ‘ f_5 ’ mean? It looks like this is the 4f case? The inset shows the relationship between the distance d and angle θ . Here $d = d_1 + d_2 + d_3$ and f_5 is the focal length of lens L5. In an ideal 4f system, the distance d should equal f_5 (300 mm here). However, at this distance, the angle θ would be 20 degrees. To decrease θ to 8 degrees, the 4f system is intentionally altered by setting d to 600 mm (by moving away the OBJ2 from PBS or increasing L distance). This adjustment effectively reduces the angle while breaking the 4f system configuration. These sentences are added to the **caption Fig. S2**.

Fig. S2: There is an arrow with the letter ‘a’ – what does this mean? We apologize for this TYPO. We have now deleted it (**Fig. S2**).

Writing:

There are a number of typos and grammatical errors throughout the manuscript that made it challenging to follow in some places. A few are noted below, but the paper should be copy edited carefully prior to publication.

Thank you for pointing this out. We have tried to go through the manuscript again and tried to fix them to the best of our ability.

“To avoid the slow translation of bulky objectives or the sample stages, several attempts, employing variable-focus (vari-focus) lenses, mechanical mirrors, and acousto-optics modulators have been proposed to refocus the light for 3D imaging.” Missing comma after ‘modulators’
It is fixed. [Line# 37].

“To the best of our knowledge, because, using purely linear optical elements, lossless conversion of unpolarized light into a single polarized state is not yet possible (Supplementary Note 1), manipulating the fluorescent light using the optical isolators is unfeasible.” – consider rewording, hard to follow.

We have now reworded it to, “To the best of our knowledge, using purely linear optical elements (like lenses, PBSs, mirrors, waveplates, etc.), lossless conversion of unpolarized light into either S or P polarization state is not yet possible^{30,31} (Supplementary Note 1). Therefore, getting the fluorescent light in and out through an optical isolator with 100 % efficiency, used in folded pmRF geometry, is not feasible.” [Line# 72-75]

“Fig. 2c shows the maximum intensity projection (MIP) of beads (from 10 axial slices, each slice spaced 500 nm) separated by 30 μm for S, P, and S+P across the scan range, after 10 iterations of Richardson-Lucy (RL) deconvolution.” – consider rewording this, it took me a long time to understand what was separated by 30 μm (groups of beads)

We apologize for this clumsily written sentence. It has been changed to: “Figure 2c displays the beads' maximum intensity projection (MIP) across the scan range. For comparison we show S, P, and combined S+P images, at the beginning (-30 μm), center (0 μm) and end (+30 μm) of the scan

range. These MIPs are generated from ten consecutive axial slices with a step size of 500 nm, using raw, unprocessed images.” [Lines#204-207]

“...both the S and P-polarized images rendered onto the camera exhibit identical FWHM” – the term ‘identical’ is confusing given that the numbers for S & P are different. Perhaps ‘comparable’, etc. would be better? Similarly in the next paragraph: “...thereby resulting in an identical ‘S+P’ image”.

We agree. This word has been replaced by “comparable” [Line#198]. The identical word is changed to similar [Line#201].

Reviewer #2 (Remarks to the Author):

Paragraph beginning: “Unlike the adaptive optics or DM-based approaches...” In this paragraph you need to address the unfolded geometry discussed by Botcherby in his original paper. You may decide to discuss the temporal limitations of the unfolded geometry (moving the entire objective mass) but you must make mention of the availability of a remote focusing configuration that can collect 100% of the light. We want to thank the reviewer for his/her comment. We have now incorporated this discussion in our main text: “It should be noted that Botcherby’s original refocusing design involving a 3rd objective could collect the entire fluorescence signal, however this design warrants translation of bulky objectives which would slow down the axial scan speed¹⁷.” [Line# 78-79] *Equally it is inaccurate to state that microscopes using pmRF carrier axial scanning necessarily lose 50% of the fluorescence light. Please change this.* We have now addressed this by rewording the sentence “As a result, microscopes that use optical isolator based pmRF to carry out axial scanning, may incur up to 50% light loss due to one state...” [Line#76]

Paragraph beginning: “Thirdly, there exists an inverse relationship between...” this paragraph states the trade-off between adhering to a 4F system or minimising the angular separation of the two polarisation images. Please state explicitly which side of this trade-off the current design adheres to.

We have now added the text: “We found that our current design still allows us to achieve resolution comparable to that of diffraction-limited systems (Fig. 1e), by compromising the 4f arrangement to minimize θ .” [Line#124-125]. We would like to mention that in our previous version, we discussed this “compromise” in detail under the section “Characterization of the optical system”.

“Characterization of the optical system” Please state why it is valid to implement the paraxial approximation for hi na lenses. The paraxial approximation is only valid and the small angle limit ($\sin(\theta) \sim \theta$). Please calculate the size of errors you would anticipate and applying the paraxial approximation to the large acceptance angles of the lenses used in this experiment. Whilst the specifics of the detection and remote objectives are given in the methods section they should first be specified in this section of the paper (including the NA and model) to enable the reader to make an informed comparison between the results of the simulation with that expected from the experiment

The reason we use ray-tracing to assess the magnification of the system is that the magnification of a system, given a certain immersion medium, is not related to the NA of the optics based on the

Abbe sine condition. We compared the magnifications calculated from the ray tracing at $NA = 0.8$ and $NA = 0.1$, the differences are in the order of 10^{-10} , therefore, even at relatively high NA, ray tracing can still give a good estimation of the system's magnification.

In addition, we want to emphasize the ray-tracing simulation was done as a design consideration. It is approximately accurate for estimating the magnification and beam size at each optical element. For Strehl ratio and PSF calculation, paraxial approximation is no longer accurate. We also measured the magnifications experimentally later to report actual numbers. Simulations using Zemax, OSLO or CODE V would have been optimal, their prohibitive cost has led us to use ray tracing as our “good enough” starting point.

Figure 1: Typo: ‘exist’ = ‘exit’

We apologize for such mistakes. It is fixed now (**Figure 1**).

As we now know, the resolution of a remote focusing system varies across the field of view. Please state explicitly where the beads analysed in the performance of your system (Figure 2) were located within the field of view of your detection objective.

The beads were analyzed within an $80 \times 15 \mu\text{m}^2$ area at the center of the light sheet, equivalent to 484×89 pixels² on the camera with a maximum FOV of 2024×2024 pixels. This central FOV represents 24% and 4% of the total FOV ‘X’ and ‘Y’ laterally. We reported the average resolution with its standard deviation for this specific FOV, providing a focused assessment of the system's performance in the most utilized imaging region.

Figure 2 caption: “Z direction exhibits less resolution in the axial direction controlled by the light sheet waist”. There seems to be a misunderstanding here. The axial extent of the fluorescent bead image is determined only by the dimensions of the bead and not the light sheet. The axial resolution would only be limited by the light sheet thickness in the case of an extended object. Please correct this part of the text.

We would like to respectfully mention that this method of obtaining PSF (moving a bead through the lightsheet to capture a 3D stack), in order to assess the lightsheet's waist, is very standard in LSFM. Numerous groups, before us, have used this idea to characterize axial resolution. However, we can understand how readers new to LSFM may get confused. Since such a discussion is beyond the scope of this manuscript, we have now provided several citations for readers to understand the concept behind the working principle of this method. **[These citations are added in Line # 210]**

Fig6 of <https://opg.optica.org/oe/fulltext.cfm?uri=oe-28-7-9464&id=429227>

Fig5 of <https://opg.optica.org/boe/fulltext.cfm?uri=boe-9-12-6154&id=401173>

Fig2 of <https://www.ncbi.nlm.nih.gov/pmc/articles/PMC4472079/>

Fig2 of <https://www.nature.com/articles/s41592-022-01417-2/figures/2>

Fig2 of <https://www.nature.com/articles/s41598-019-53875-y>

Fig1 & SupFig2 of <https://www.nature.com/articles/s41592-019-0615-4>

Fig1b of <https://www.nature.com/articles/s41592-019-0615-4>

Figure 2: “The minimum lateral resolution, 394 nm, occurs at the center of the scan range”. This statement is true for the Y direction but not true for the X direction where the minimum is closer to 15 microns in Z. Please provide a justification for why this might be.

We removed “minimum” and changed the sentence to “The figure shows a lateral FWHM of 570 ± 26 nm and 666 ± 44 nm in X and Y directions at the center of the scan range respectively” **[Line#225-227]**

Figure 2b. This figure shows the Z location in terms of the galvo position. As the reader has no way of mapping between the 'galvo' position (light sheet) to the 'Z' position (mirror) please redraw Fig 2B in terms of the Z units for consistency.

Figure 2b is fixed now. Z location ranges from -40 μm to 40 μm .

Deconvolution: Deconvolution algorithms have no place in this work. Richardson Lucy deconvolution actively de-blurs the raw bead images, changing completely what it is that you are reporting. Please redraw Figure 2d for the raw bead image data to provide a true reflection of the system performance.

We thank the reviewer for this comment and wholeheartedly agree with his/her assessment. We have now redrawn our figures showing raw and unprocessed images.

"In the axial direction (the YZ view) the PSFs are limited by the Gaussian light sheet's waist". Please see earlier comment.

See the previous answer.

Discussion

Paragraph beginning: "Although recent advancements in single-objective oblique plane microscopy (OPM) have achieved speeds comparable to our method". It is important to acknowledge the advantages of a single objective like sheet methods compared to orthogonal light sheet methods in terms of sample accessibility. Many imaging experiments in biomedical science are not possible using the orthogonal objective like sheet method.

We thank the reviewer for this comment and want to emphasize that the very reason we chose to discuss this modality in our paper is because of the unique advantages OPM presents in terms of 3D volumetric imaging and therefore its rapid gain in popularity. However, we can understand if this was not enough. We have now incorporated the following text "Although recent advancements in oblique plane microscopy (OPM), which incorporates the benefits of LSMs to the convenience of single-objective microscopes, have achieved speeds comparable to our method, our technique presents several notable advantages." [Line# 313-315]

In the new configuration it is not possible to inject the excitation light via the polarising beam splitter. Please comment on how the new detection configuration would integrate other optical sectioning technologies; specifically spinning disc, single objective light sheet, 2 photon and structured illumination.

We disagree with this assessment. It is possible to inject the laser through the same path the fluorescence came through since the linearly polarized laser can easily be split in S and P follow the same path and get combined at the PBS. We have provided an optical schematic for a 2-photon design here.

Hoping that the reviewer (after seeing the schematic) is convinced that laser injection is indeed possible, the reviewer can easily see that our scheme is indeed possible in spinning disc, 2-photon, and structured illumination. With that said we are not incorporating any commentary in our main manuscript regarding this, *since we believe that our method's utility is more widespread and is not limited to the list provided by the reviewer.*

Editorial Note: Figure redacted

Regarding the single objective lightsheet: we are a bit unsure why the reviewer asked us to show the utility of our de-scanning concept in a single-objective-light-sheet (OPM). Single-objective-light sheets typically do not need axial de-scanning. In OPM, the “de-scanning” is done through the scanning galvo which moves the lightsheet in the lateral direction. With that said, even for certain single objective lightsheet designs we can also use our polarization splitting/combining idea. We will explore these in our future publications.

Please comment on the feasibility of injecting excitation light into the ‘detection’ objective via a dichroic mirror to enable single objective light sheet illumination and polarisation insensitive fluorescence detection.

Please see the above comment. The drawing is provided as a separate sheet.

*Methods: Optical setup: It is not clear to me why you have chosen objectives with different magnifications. In order for the remote focusing principles to apply, the only magnification of the stigmatic remote image is due to the index difference between the water sample and the air immersion mirror (i.e. a magnification of 4/3). While the reviewer is correct in his/her understanding of overall magnification in RF, the reviewer is incorrect regarding his/her assessment of using different magnification objectives. Botcherby in his very first RF paper prescribed the recipe to get the desired overall magnification by employing different magnification objectives. We want to emphasize that coupling different magnification objectives is a very standard practice in RF and humbly suggest the reviewer to consult Botcherby's paper on how to pupil-match different magnification objectives [Botcherby, et al., *Optics letters* 32.14 (2007)]. *By using a 40X imaging objective coupled to a 20X remote objective you introduce an extra factor of 2 into the magnification.* This is an incorrect assessment. Our overall magnification ratio is still*

4/3. It's important to note that the two objectives, labeled obj1 and obj2, are pupil-matched by lenses with focal lengths of 200 mm and 300 mm, respectively. These focal lengths are critical for achieving proper pupil matching between the lenses. As a result, the second objective provides a magnification of 30x, contrary to the expected 20x (because we used 300 mm, not 200 mm). Therefore, with the first microscope objective magnifying by 40x and the second de-magnifying by 30x, we arrive at a net magnification of 4/3.

References: many of the references are poorly formatted some with author names missing completely. Please reformat reference list.

We are extremely sorry for this. References are fixed.

Reviewer #3 (Remarks to the Author):

In the manuscript “Axial de-scanning using remote refocusing in the detection arm of light-sheet fluorescence microscopy”, Dibaji et al. present a novel optical configuration for minimizing light loss in a folded aberration-free remote refocusing relay. In the presented design, the collected fluorescence, split into s- and p-polarizations by a polarized beam splitter, is reflected into the pupil of the second objective in the remote-refocusing relay. The light is back-reflected along the same path using a mirror perpendicular the optical axis of Obj2, reversing the polarization and then coupled out of the remote refocusing relay by the PBS, recombined at the detector. This approach allows the use of light with the polarization that is typically lost in a folded remote refocusing relay, improving the optical collection efficiency. The method proposed in the manuscript is a clever and original solution to the main trade-off when using folded remote-refocusing in the detection path of a fluorescence microscope, and although the implementation involves its own trade-off, it may be a useful solution for certain applications.

The manuscript presents a detailed characterization of the system design and performance, as well as some proof-of-concept applications, and discussion of the trade-offs. The manuscript is well-written with good quality data presentation and figures, and would be of interest to a wide audience within the microscopy development community. We deeply appreciate the reviewer's agreement with our sentiment. Although our initial proof-of-concept study has areas for enhancement, the core concept holds the potential to revolutionize various aspects of microscopy. I believe the manuscript would be suitable for publication in this journal, with some minor revisions.

Main points:

• *On the second page, the authors state that “in the detection arm, the fluorescence is unpolarized in nature”. This is not always entirely true – for fluorophores with high levels of fluorescence anisotropy and polarized excitation light, the fluorescence can be partially polarized, with less than 50% loss during PBS double pass. The manuscript would benefit from adding a brief discussion of the role of fluorescence anisotropy on light loss through the polarized beam splitter when remote-refocusing in the detection path. In this case transmission can be maximized using a HWP before the PBS (see Sparks et al (2020), DOI: 10.1002/jbio.201960239).*

We thank the reviewer for bringing this point. We agree with the reviewer that it is important to acknowledge exceptions to this (fluorescence is unpolarized), particularly in the case of

fluorophores exhibiting significant fluorescence anisotropy especially when polarized excitation light is used. In such scenarios, the emitted fluorescence can retain a degree of polarization, potentially leading to less than a 50% loss during pass through a polarizing beam splitter (PBS). We appreciate the suggestion to discuss the impact of fluorescence anisotropy on light loss during remote refocusing in the detection path. Indeed, the transmission efficiency in such cases can be optimized by incorporating a half-wave plate (HWP) before the PBS, as demonstrated by Sparks et al. (2020).

With that said we would like to point out that our scheme can still be useful for anisotropic fluorescence and automatically redistributes the fluorescence light in either arm. We have included a section discussing this aspect to provide a comprehensive understanding of the fluorescence behavior in our system under varying conditions. [Line#71-77]

Reworded as follows:

In the detection arm, however, due to the fluorescence anisotropy²⁵, the emitted fluorescence may be partially polarized in nature. To the best of our knowledge, using purely linear optical elements (like lenses, PBSs, mirrors, waveplates, etc.), lossless conversion of unpolarized light into either S or P polarization state is not yet possible^{30,31} (Supplementary Note 1). Therefore, achieving 100% efficiency in transmitting fluorescent light in and out through an optical isolator, which is used in folded pmRF geometry, is not feasible. As a result, microscopes that use optical isolators based pmRF to carry out axial scanning may incur up to 50% light loss due to one state of the polarized light being discarded after the PBS^{17,22,25} (Supplementary Fig. 1a).

• The authors state that in order to minimize the loss of FOV in the proposed configuration, the design benefits from a small angle θ , as this minimizes the separation of the two remote images formed by Obj2, enabling the use of a smaller mirror for refocusing and reducing the loss of FOV.

o I would rephrase the last sentence which states a “smaller ΔL ...guarantees that both images fit within Obj2 FOV” – as this does not guarantee this. Even for a very small L, Obj2 can limit overall FOV, depending on objective choice.

We thank the reviewer for pointing this out and agree with him/her that the use of the word “guarantee” is unnecessarily assertive and therefore could be misleading. We have now changed this to “for a particular objective the two images are more towards the center of the FOV, which may help reduce field dependent aberrations and improve the collection efficiency.” [Line 117-118]

o The authors propose breaking the 4f system arrangement to increase the distance between Obj2 and the PBS, reducing θ , and that the compromise in the current design still achieves diffraction-limited (DL) resolution. However, it is not clear from the manuscript as to what the DL remote refocusing range would be when maintaining the pupil-matched 4f remote refocusing configuration as opposed to minimizing θ . As the DL range is reduced, image contrast away from the nominal focal plane is also reduced – which may counteract the benefit of improving the collection efficiency in folded remote refocusing in detection using the configuration proposed by the authors.

We thank the reviewer for pointing out this insightful comment. However, we want to respectfully mention that we were aware of this and discussed this shortcoming in our discussion section (2nd

paragraph, last three lines). We would like the Reviewer to kindly consider these two critical points while making the assessment:

1. We agree that in our current design, we don't make use of the entire DL range that RF can offer at this FN and NA, however, we feel that our 70 μm axial scan range may be sufficient in many cases (one example we show in our live cell imaging).
2. The steep angle proposed in our design *is not a fundamental problem*. As the Reviewer correctly mentions previously, this work is a proof-of-concept where we feel the most important aspect of our work is the realization to distribute the unpolarized/partially-polarized light in S and P and then keep the S and P paths always separate and be able to combine them using a single PBS. Future designs using off-the-shelf components can very easily reduce the angle, thereby bringing the 2nd objective to a perfect 4f configuration. Of course, a more eloquent solution will be to involve companies like ASI/Thorlabs to fabricate PBS+mirror assembly that can make a more compact unit which will reduce such angular constraint further.

o The manuscript would benefit from including a more detailed discussion of the trade-off between FOV, collection efficiency through the PBS, and DL range of the remote refocusing at various NA, as this would be the key consideration when deciding whether the proposed configuration would be useful. However, I don't think that a quantitative analysis is necessarily required within the scope of the manuscript which is more focused on a proof-of-concept demonstration of the principle.

We sincerely want to thank the reviewer for bringing this point up. The collection efficiency aspect needs to be explored in more detail and quantitatively (both theoretically and experimentally) as this can result in better design considerations. Here we followed the Reviewer's suggestion and did a trigonometric analysis inspired by a previous work and added a paragraph in the discussion section.

"Following the footsteps of Hong et al.⁴⁵ we considered how our off-axis remote-focusing design might affect the collection efficiency^{46,47} and therefore the achievable DL range. As it turns out, because in our current design the two images after the Obj2 are formed at the far end of the FOV, some of the rays always miss the Obj2 when we try to defocus. Our trigonometric analysis suggests that, assuming a 90% collection efficiency as the acceptable limit, the scan range allowed should have been 253 μm (based on collection efficiency alone). However, we know that is not the case: as of now our DL range is limited by the mismatched 4f which limits it to only 70 μm . Based on this analysis, it is imperative that to get the most out of the DL range and the collection efficiency of the remote-focusing system, while designing, the goal should be to not only maintain a strict 4f condition between L1 and L2 but also reduce the ΔL . As such we found that for a particular NA, having a lower magnification on the remote-focusing objective has two intertwined advantages: (1) it allows the user to choose a longer L2 which reduces the angular burden on the S and P paths (2) assuming a similar field number (FN), lower magnification objectives have larger FOV (=FN/Magnification) which reduces the requirement of smaller ΔL to begin with. Our future designs will take these aspects into consideration and investigate newer designs which aim to reduce the angle between the S and P beam while adhering to the 4f condition between L1 and L2." [line#289-302]

o Using the deconvolved bead volumes to evaluate the DL range can be misleading as the quoted axial resolution of deconvolved beads may be subject to the deconvolution parameters and not

fully representative of the raw optical performance of the system. I would propose using the raw bead data, and accounting for bead size and pixels size to estimate the PSF FWHM.

We wholeheartedly agree with the reviewer on this. For these reasons we have now redrawn the figures using raw, unprocessed images and depict resolution numbers based on that.

Minor comments:

• Page 2, Line 2 – The authors state that “for an objective with an NA of 0.8, the maximum axial scan range that DM based technique can generate is approximately 40 μm ”. However, in work by Wright et al (2021), (DOI: 10.1088/2515-7647/ac29a2), a 100 μm scan range is achieved with a 0.8 NA lens (with a Strehl ratio of >0.6 over 77 micron refocus range). A similar scan range is achieved in a LSFM implementation with a slightly lower NA by Hong et al (2023) (DOI: 10.1063/5.0125946)

We thank the reviewer for pointing out the recent advancements in axial scan ranges with similar NA objectives. The discrepancy in the reported scan range in our manuscript compared to that achieved by Wright et al. (2021) and Hong et al. (2023) underscores the rapid progress in the field. We will update our manuscript to reflect a 100 μm scan range is indeed possible as demonstrated in these studies. **[Line#52-53]**

• Results, second paragraph, Line 2 “M3 consistently moves in parallel with the focal plane of Obj2” – I think it might be more clear to state that M3 translates along the optical axis of O2, or perpendicular to the nominal focal plane of O2.

“M3 consistently translates along the optical axis of Obj2 without any angular deviation” is used. **[Line #108].**

• Last line of page 3 (“The small deviation is limited by the geometry...”). Referring to a magnification mismatch of 0.07 (5%) as “small” is rather subjective for a system with the current NA. Mohanan and Corbett (2020, 2023) and Hong et al (2023) present a detailed characterization of remote refocusing performance and impact of magnification mismatch on the diffraction-limited refocusing range at various NA systems, suggesting that even a 1% mismatch can reduce the DL range to half its maximum value.

We have now removed the word “small” and simply report the 5% magnification mismatch. **[Line#172]**

• In Supplementary Figure 9, describing magnification calibration, there appears to be some minor clock-wise rotation of the calibration target relative to the FOV x-y orientation. Is this accounted for when estimating the peak separation of the features?

We thank the reviewer for pointing this out. We estimated the rotation angle of the calibration target to be 16.4 mrad, which will result in 0.0134% error in the magnification estimation. As the error is small, therefore we didn't account for the rotation in the magnification estimation.

• 4th paragraph in Discussion: “ The OPM setup necessitates a third objective...” – this is not strictly true – OPM can also be implemented using folded remote refocusing with a tilted mirror in remote refocus space – see the dOPM setup by Sparks et al (2020) (DOI: 10.1364/BOE.409781)

We agree and are aware of other works like Kim et al., (<https://doi.org/10.1038/s41592-019-0510-z>). However here we simply wanted to say that recent OPM setups employing a 3rd objective have

shown improved light-collectability by using expensive objectives like ‘Snouty’ or ‘King Snout’. However we understand that our previous wording may insinuate that all OPMs require a 3rd objective. He have now reworded this. [Line# 321-322]

Grammar/formatting:

- *Line 2 of main text – “BioPhotonics” – no need for capitalization at start or middle of word*
It is fixed. biophotonics [Line#30]

- *Two sentences on page 2 - not sure whether the use of italics is deliberate, but is perhaps not necessary.*
The italics sentences were changed to non-italics.

- *Line 6 of discussion “Carryout” – 2 words*
It is fixed [Line#275]

- *Line 3 of Results section - “Botceherby” – Botcherby*
It is fixed [Line# 94]

REVIEWER COMMENTS

Reviewer #1 (Remarks to the Author):

I thank the authors for considering my comments and for their thoughtful responses. The majority of my concerns have been addressed. A few minor remaining comments are below – once these are addressed, I believe the paper will be ready for publication. This is a nice manuscript and I look forward to sharing the published version in a future journal club.

Thank you for clarifying the 8 deg angle, square vs. rectangular pixels, and PSF modeling methods – all make sense now. I'll also note that the writing clarity is much improved, and I appreciate the time the authors took here – I believe this will help improve the reach of this technique.

Remaining comments:

Typo line 449: “Here, the coverslips were faced 8 degrees relative to the optic axis of the detection objective (Supplementary Fig.7)” – I think this should reference Supplementary Fig. 8 instead (showing the 8 deg coverslip angle).

Regarding the slit width (line 487): the explanation provided by the authors in the response document is helpful (i.e. that this is a physical slit in the illumination light path, which is used to adjust the illumination NA). Because the term ‘slit width’ is also used with different meanings in related contexts (e.g. to refer to the number of pixels in a CMOS camera’s rolling shutter for axially swept light-sheet microscopy, or for the confocal slit width in line-scanned confocal microscopy), I recommend adding this explanation to the manuscript text.

It seems like the naming of lenses in Fig. 1, Fig. S2, and the main text still has a number of errors/inconsistencies. For example, the text mentions L1 and L2, which do not appear in Fig. 1 anywhere. The figure includes L6, which is not mentioned anywhere else. The text mentions the “Tube lens” in multiple places without specifying which lens this is: is it L6 (the lens in a tube lens position) or one of the relay lenses (an actual TTL tube lens from Thorlabs)? Please carefully double check all the naming conventions in the figure and text – I think errors here are very likely to confuse readers, especially those new to light sheet/remote focusing.

Reviewer #2 (Remarks to the Author):

Thank you for the thoughtful replies to the reviewer comments. I have looked through all of the responses and the revised manuscript and am happy for the manuscript to be published. I have only one outstanding comment, which is that the corrections which claim to have been made to Fig 2b, do not appear to be in evidence in the latest draft of the manuscript (i.e. the x-axis still reads as 'galvo position' and not 'Z location').

Reviewer #3 (Remarks to the Author):

In the revised version of the “Axial de-scanning using remote focusing in the detection arm of light-sheet microscopy” manuscript, the authors have adequately addressed my concerns presented in first round of reviews including adding a discussion of the role of fluorescence anisotropy and trade-offs between the diffraction limited range and collection efficiency, resolution characterization, references to more recent advances in remote refocusing, as well as more minor style and language/phrasing comments. I think that, subject to further minor revisions and optional suggestions, the manuscript is suitable for publication in this journal and would be of interest to the wider microscopy development community.

Here are some further minor comments:

- Line 156 – “simple lenses” – Did you mean “thin lenses” i.e. of zero thickness and satisfying the thin lens approximation? “Simple” lenses usually refer to singlets.
- Line 497 – Dipimage toolbox – is this missing a reference?
- Lines 461-462, 492-493 and Supplementary Fig 4 (Quantification of light-sheet dimensions). Strictly speaking, the light-sheet dimensions (beam waist and Rayleigh length) are not given directly by FWHM_z of the bead images. You can get the optical sectioning (and hence light-sheet thickness ω) from the FWHM of the axial profile of the laterally integrated intensity of a point source (See Sheppard & Wilson, 1978 and Wilson 1989) – this method has been used to characterize optical sectioning in for example Ref 25). While you can still use the propagation distance over which FWHM_z doubles as an indirect metric of FOV, I would remove the part of the caption of Supplementary Figure 2 which implies equivalence of FWHM_z and ω .
- Some of the reference formatting is off: Some journal article references do not list authors at all (Refs 24, 27, 40 etc) and some seem incomplete (including Ref 54 etc).

Further suggestions and potential points for discussion:

- In Figure 1 b-c-d, I think that colour coding the two emission light paths would potentially help with clarity (as is done in the inset of Supplementary Figure 1c)

- Figure 2c – the bead images at the -30 μm position appear to be vertically elongated, round at 0 defocus and horizontally stretched at the +30 μm position, particularly for the S polarization. Do you know whether, in this particular case, this is a result of misalignment/optical aberrations in that particular refocusing path, whether it is something to do with sample geometry relative to detection objective, or whether this is an inherent polarization effect?

- Supplementary Fig. 8 – Re: 8 degree angle between coverslip and detection plane – is that sufficient to not significantly obscure or aberrate the detection collection cone for 0.8 NA? Given that the full illumination NA was not used (the authors mention that they prioritized FOV and hence used a tighter slit width to get a longer Rayleigh length), would a sample orientation with the coverslip orthogonal or at a small angle to the focal plane of the detection objective be preferable (i.e with detection & illumination objective geometry relative to coverslip swapped around)?

- In Figure 2d for FWHM_x and FWHM_y, the vertical axis range could be adjusted to 0.4-1 μm (instead of 0.2-1.2 μm) as otherwise is it difficult to assess changes with depth and minor differences between S, P and S+P. I don't think there is a need to have the exact same vertical axis range for the lateral and axial FWHM.

Reviewers' comments:

Reviewer #1 (Remarks to the Author):

I thank the authors for considering my comments and for their thoughtful responses. The majority of my concerns have been addressed. A few minor remaining comments are below – once these are addressed, I believe the paper will be ready for publication. This is a nice manuscript and I look forward to sharing the published version in a future journal club.

Thank you for clarifying the 8 deg angle, square vs. rectangular pixels, and PSF modeling methods – all make sense now. I'll also note that the writing clarity is much improved, and I appreciate the time the authors took here – I believe this will help improve the reach of this technique.

Remaining comments:

Typo line 449: “Here, the coverslips were faced 8 degrees relative to the optic axis of the detection objective (Supplementary Fig.7)” – I think this should reference Supplementary Fig. 8 instead (showing the 8 deg coverslip angle). We agree and apologize for this mistake. We have now updated the reference to Supplementary Fig. 8. (Line # 455)

Regarding the slit width (line 487): the explanation provided by the authors in the response document is helpful (i.e. that this is a physical slit in the illumination light path, which is used to adjust the illumination NA). Because the term ‘slit width’ is also used with different meanings in related contexts (e.g. to refer to the number of pixels in a CMOS camera’s rolling shutter for axially swept light-sheet microscopy, or for the confocal slit width in line-scanned confocal microscopy), I recommend adding this explanation to the manuscript text. We thank the reviewer’s comment, and we added this explanation to lines #493-494.

It seems like the naming of lenses in Fig. 1, Fig. S2, and the main text still has a number of errors/inconsistencies. For example, the text mentions L1 and L2, which do not appear in Fig. 1 anywhere. The figure includes L6, which is not mentioned anywhere else. The text mentions the “Tube lens” in multiple places without specifying which lens this is: is it L6 (the lens in a tube lens position) or one of the relay lenses (an actual TTL tube lens from Thorlabs)? Please carefully double check all the naming conventions in the figure and text – I think errors here are very likely to confuse readers, especially those new to light sheet/remote focusing. We thank the reviewer’s comments, we made the following changes:

Line #104: tube lens to: lens (L6)

Line # 121: lenses L1 and L2 to: lens L4 and lens 5

Line # 169: detection objective to the tube lens to: detection objective (Obj1) to the tube lens (L4)

Line # 170: tube lens to: tube lens (L4)

Line #183-184: tube lens to: lens (L6)

Line # 297: L1 and L2 to: L4 and L5

Line # 299: a longer L2 to: a longer L5

Line# 302: L1 and L2 to: L4 and L5

Line #380: tube lens (TTL200-A, Thorlabs) to: tube lens (L4) (TTL200-A, Thorlabs)
Line #531 tube lens and the detection objective to: tube lens (L4) and the detection objective (Obj1)
Fig S1 is updated.

Reviewer #2 (Remarks to the Author):

Thank you for the thoughtful replies to the reviewer comments. I have looked through all of the responses and the revised manuscript and am happy for the manuscript to be published.

I have only one outstanding comment, which is that the corrections which claim to have been made to Fig 2b, do not appear to be in evidence in the latest draft of the manuscript (i.e. the x-axis still reads as 'galvo position' and not 'Z location'). We apologize to the reviewer for this slip-up. We have now updated the x-axis for Fig. 2b.

Reviewer #3 (Remarks to the Author):

In the revised version of the “Axial de-scanning using remote focusing in the detection arm of light-sheet microscopy” manuscript, the authors have adequately addressed my concerns presented in first round of reviews including adding a discussion of the role of fluorescence anisotropy and trade-offs between the diffraction limited range and collection efficiency, resolution characterization, references to more recent advances in remote refocusing, as well as more minor style and language/phrasing comments. I think that, subject to further minor revisions and optional suggestions, the manuscript is suitable for publication in this journal and would be of interest to the wider microscopy development community.

Here are some further minor comments:

- Line 156 – “simple lenses” – Did you mean “thin lenses” i.e. of zero thickness and satisfying the thin lens approximation? “Simple” lenses usually refer to singlets. We thank the reviewer’s comment, we changed “simple lenses” to “thin lenses” (Line #156)

- Line 497 – Dipimage toolbox – is this missing a reference? We thank the reviewer’s comments, we added the reference to Dipimage toolbox.

- Lines 461-462, 492-493 and Supplementary Fig 4 (Quantification of light-sheet dimensions). Strictly speaking, the light-sheet dimensions (beam waist and Rayleigh length) are not given directly by $FWHM_z$ of the bead images. You can get the optical sectioning (and hence light-sheet thickness ω) from the FWHM of the axial profile of the laterally integrated intensity of a point source (See Sheppard & Wilson, 1978 and Wilson 1989) – this method has been used to characterize optical sectioning in for example Ref 25). While you can still use the propagation distance over which $FWHM_z$ doubles as an indirect metric of FOV, I would remove the part of the caption of Supplementary Figure 2 which implies equivalence of $FWHM_z$ and ω . We agree with the reviewer, it is true that the $FWHM_z$ of the bead image is not equivalent to the waist of the light-sheet waist, although often used as such. Nonetheless, we updated the main text and the

caption for Supplementary Fig. 4 to indicate that we only use the FWHMz to define the FOV of the light-sheet.

- *Some of the reference formatting is off: Some journal article references do not list authors at all (Refs 24, 27, 40 etc) and some seem incomplete (including Ref 54 etc).* We thought we updated the references specifically but may have gotten overwritten by our reference manager. We sincerely apologize for this oversight. Anyways, we have now updated the reference format.

Further suggestions and potential points for discussion:

- *In Figure 1 b-c-d, I think that colour coding the two emission light paths would potentially help with clarity (as is done in the inset of Supplementary Figure 1c)* We want to thank the reviewer for this suggestion. We want to emphasize that our earlier figures about this schematic indeed had S and P as color coded. However, we found, through personal experiences, that this color coding seems to confuse the audience. Often it is perceived that we are splitting the fluorescence into two wavelengths and then adding them. For these reasons we have decided to keep the existing formatting.

- *Figure 2c – the bead images at the $-30\ \mu\text{m}$ position appear to be vertically elongated, round at 0 defocus and horizontally stretched at the $+30\ \mu\text{m}$ position, particularly for the S polarization. Do you know whether, in this particular case, this is a result of misalignment/optical aberrations in that particular refocusing path, whether it is something to do with sample geometry relative to detection objective, or whether this is an inherent polarization effect?* We want to thank the reviewer for this comment; however, we would like to point out that we already discussed these in our original manuscript (line 227-232). As mentioned, we found that both S and P suffered from this problem (uneven elongation at $\pm 30\ \mu\text{m}$). Although we haven't particularly investigated this depth-dependent astigmatism, we feel this aberration could be coming from the two intertwined sources: (1) the two foci after the remote-focusing objective are being reflected from the edge of the FOV (1.42 mm separated), (2) in our existing design the remote-focusing objective doesn't satisfy the 4f condition. Our future iterations will try to address both issues.

- *Supplementary Fig. 8 – Re: 8 degree angle between coverslip and detection plane – is that sufficient to not significantly obscure or aberrate the detection collection cone for 0.8 NA? Given that the full illumination NA was not used (the authors mention that they prioritized FOV and hence used a tighter slit width to get a longer Rayleigh length), would a sample orientation with the coverslip orthogonal or at a small angle to the focal plane of the detection objective be preferable (i.e with detection & illumination objective geometry relative to coverslip swapped around)?* The reviewer is correct in his/her assessment. It is true that there will be a portion of the fluorescent light passing through the coverslip. With the current 8-degree tilt, we estimate that about 25% of light will go through the coverslip and will form a blurred background image. To avoid this, the coverslip needs to be tilted by at least the maximum collection angle of the detection objective. However, at large tilt angle, the light-sheet needs to be scanned both along Z and Y axes (<https://doi.org/10.1016/j.bpj.2016.01.029>). For this proof-of-concept study, our motivation was focused on the detection side, we used a relatively simple excitation path to translate the light-

sheet at a small tilt angle. In future design, we will implement a more sophisticated excitation path to translate the light-sheet at large tilt angles.

- *In Figure 2d for FWHM_x and FWHM_y, the vertical axis range could be adjusted to 0.4-1 μm (instead of 0.2-1.2 μm) as otherwise is it difficult to assess changes with depth and minor differences between S, P and S+P. I don't think there is a need to have the exact same vertical axis range for the lateral and axial FWHM.* We updated the vertical axis range to 0.4-1.2 μm, we prefer to keep the axis range consistent between all plots, so that the readers could easily see the resolution difference between the lateral and axial dimensions.

REVIEWERS' COMMENTS

Reviewer #3 (Remarks to the Author):

Thank you to the authors for the extensive work addressing the comments and concerns raised by me and other reviewers in the first and second review rounds. I think the manuscript is suitable for publication and I don't have any further concerns to raise.